



# A Chemical Ionization Mass Spectrometry Utilizing Ammonium Ions ($NH_4^+$ CIMS) for Measurements of Organic Compounds in the Atmosphere

Lu Xu[1,2], Matthew M. Coggon[2], Chelsea E. Stockwell[1,2], Jessica B. Gilman[2], Michael A. Robinson[1,2,3], Martin Breitenlechner[1,2], Aaron Lamplugh[1,2,*], J. Andrew Neuman[1,2], Gordon A. Novak[1,2], Patrick R. Veres[2], Steven S. Brown[2,3], and Carsten Warneke[2]

[1]Cooperative Institute for Research in Environmental Sciences, University of Colorado Boulder, Boulder, Colorado, USA
[2]NOAA Chemical Sciences Laboratory, Boulder, Colorado, USA
[3]Department of Chemistry, University of Colorado Boulder, Boulder, Colorado, USA
[*]Now at Institute of Behavioral Science, University of Colorado Boulder, Boulder, Colorado, USA

**Correspondence:** Lu Xu (lu.xu@noaa.gov) and Carsten Warneke (carsten.warneke@noaa.gov)

**Abstract.** We describe the characterization and field deployment of a Chemical Ionization Mass Spectrometry (CIMS) using a recently developed focusing ion-molecule reactor (FIMR) and ammonium-water cluster ($NH_4^+ \cdot H_2O$) as the reagent ion (denoted as $NH_4^+$ CIMS). We show that $NH_4^+ \cdot H_2O$ is a highly versatile reagent ion for measurements of a wide range of oxygenated organic compounds. The major product ion is the cluster with $NH_4^+$ produced via ligand-switching reactions. Other product ions

(e.g., protonated ion, cluster ion with $NH_4^+ \cdot H_2O$, with $H_3O^+$, and with $H_3O^+ \cdot H_2O$) are also produced, but with minor fractions for most of the oxygenated compounds studied here. The instrument sensitivities (counts per second per ppbv, cps ppbv$^{-1}$) and product distributions are strongly dependent on the instrument operating conditions, including the ratio of ammonia ($NH_3$) and $H_2O$ flows and the drift voltages, which should be carefully selected to ensure $NH_4^+ \cdot H_2O$ as the predominant reagent ion and to optimize sensitivities. For monofunctional analytes, the $NH_4^+ \cdot H_2O$ chemistry exhibits high sensitivity (i.e., > 1000 cps ppbv$^{-1}$)

towards ketones, moderate sensitivity (i.e., between 100 and 1000 cps ppbv$^{-1}$) towards aldehdyes, alcohols, organic acids, and monoterpenes, low sensitivity (i.e., between 10 and 100 cps ppbv$^{-1}$) towards isoprene and C1 and C2 organics, and negligible sensitivity (i.e., < 10 cps ppbv$^{-1}$) towards reduced aromatics. The instrumental sensitivities of analytes depend on the binding energy of the analyte-$NH_4^+$ cluster, which can be estimated using voltage scanning. This offers the possibility to constrain the sensitivity of analytes for which no calibration standards exist. This instrument was deployed in the RECAP campaign

(Re-Evaluating the Chemistry of Air Pollutants in California) in Pasadena, California during summer 2021. Measurement comparisons against co-located mass spectrometers show that the $NH_4^+$ CIMS is capable of detecting compounds from a wide range of chemical classes. The $NH_4^+$ CIMS is valuable for quantification of oxygenated VOCs and is complementary to existing chemical ionization schemes.





## 1 Introduction

Quantifying atmospheric volatile organic compounds (VOCs) and their oxidation products is critical for understanding the formation of ozone ($O_3$) and organic aerosol (OA). However, this objective has been a longstanding challenge because of the sheer number and significant chemical complexity of organic compounds in the atmosphere (Goldstein and Galbally, 2007). Chemical ionization mass spectrometry (CIMS) is a widely used and rapidly developing technique to characterize atmospheric trace gases. The advantages of CIMS include fast time response, high selectivity and sensitivity, and detection linearity over a

wide range of analyate mixing ratios. In CIMS, the analytes are ionized via ion-molecule reactions with a reagent ion, which is soft and largely preserves the identity of the analytes. The detection capability of CIMS depends on the selection of reagent ions, which are sensitive towards different classes of organics. The commonly employed reagent ions include $H_3O^+$ to detect reduced and small functionalized VOCs (de Gouw and Warneke, 2007), $I^-$ to detect inorganics and polar and acidic organics (Lee et al., 2014; Robinson et al., 2022), $CF_3O^-$ to detect organic peroxides and other multifunctional organics (Crounse

et al., 2006; Xu et al., 2020), $SF_6^-$ to detect organic acids (Nah et al., 2018), $NO_3^-$ to detect highly oxygenated molecules (Ehn et al., 2014), and protonated amines to detect reactive radicals (Berndt et al., 2018). Exploring novel reagent ions is an active research area to expand the detection capability of CIMS and to provide precise measurements of atmospheric species with high sensitivity. These efforts enable a comprehensive description of the complex mixture of atmospheric organic compounds.

One ionization scheme under active development utilizes the ammonium ion ($NH_4^+$) chemistry. Several recent studies have

demonstrated its capability to detect a range of oxygenated organic compounds, including alcohols, aldehydes, ketones, and even the short-lived peroxy radicals ($RO_2$) (Canaval et al., 2019; Hansel et al., 2018; Müller et al., 2020; Zaytsev et al., 2019; Berndt et al., 2018; Khare et al., 2022). One reason $NH_4^+$ chemistry is attractive is that it detects oxygenated organic compounds in the positive mass spectrometer mode, in contrast to existing reagent ions (i.e., $I^-$, $CF_3O^-$, and $NO_3^-$) which are operated in negative mode. This offers the potential to rapidly switch between $NH_4^+$ and $H_3O^+$ within the same instrument

to detect both oxygenated and reduced organic compounds, respectively, without substantial alteration of the electric fields in the mass spectrometer. Zaytsev et al. (2019) and Müller et al. (2020) demonstrated the feasibility of such rapid switching in laboratory conditions. The application of $NH_4^+$ CIMS in recent studies has largely focused on laboratory studies (Berndt et al., 2018; Zaytsev et al., 2019), but its deployment in field measurements and inter-comparison with other analytical instruments are scarce (Khare et al., 2022).

The instrument design, including the ion source and the ion-molecule reactor (IMR), differs between studies. Hansel et al. (2018) applied $NH_4^+$ ion chemistry in a PTR3 instrument (Breitenlechner et al., 2017) (i.e., $NH_4^+ - PTR3$) and detected peroxy radicals and other products from cyclohexene ozonolysis with sensitivities up to 28 cps ppt$^{-1}$ (ion counts per second per part per trillion by volume) in a free-jet flow system. Using a similar instrument, Zaytsev et al. (2019) calibrated 16 compounds, with a maximum sensitivity of 89 cps ppt$^{-1}$ for decanone. In both studies, the major reagent ion is $NH_4^+ \cdot H_2O$, generated

in a corona discharge ion source from a mixture of $NH_3$ and $H_2O$ gas. Later, Müller et al. (2020) developed a method to produce $NH_4^+$ using a mixture of water vapor and nitrogen in a hollow cathode glow discharge ion source, which is used in PTR-MS instruments with a traditional drift tube design that includes extraction plates between the hollow cathode ion source





and drift tube. Canaval et al. (2019) used a Selective Reagent Ionization Time-of-Flight Mass Spectrometer (SRI-ToF-MS) to produce $NH_4^+$ via reaction of $He^+$ and gas $NH_3$. Different instrument designs affect the distribution of reagent ions (i.e., $NH_4^+$

vs $NH_4^+ \cdot H_2O$ vs $NH_4^+ \cdot NH_3$), detection efficiency, and sensitivity.

In this study, we describe the performance of a $NH_4^+$ CIMS using a Tofwerk Vocus long Time-of-Flight Mass Spectrometer (Krechmer et al., 2018). We investigate the impacts of instrument conditions on the distribution of reagent ions and the instrumental sensitivities of 60 analytes from several chemical functional classes. Building upon extensive calibrations, we explore the dependence of sensitivity on the ion-molecule reaction rate constant and the binding energy of analyte-$NH_4^+$ cluster, aiming

to derive a relationship to approximate the sensitivity of analytes for which no calibration standards exist. Further, this instrument was deployed during the RECAP campaign (Re-Evaluating the Chemistry of Air Pollutants in California) in Pasadena, California during the summer of 2021. The instrument performance is further evaluated by comparison to several co-located mass spectrometers.

## 2    Experimental Methods

### 65    2.1    Instrument Description

The instrument in this work is based on the Tofwer Vocus, which utilizes a new ion source, a focusing ion-molecule reactor (FIMR), and a long Time-of-Flight Mass Spectrometer (LToF). A detailed description of the Vocus can be found in Krechmer et al. (2018). Here we briefly summarize the generation of reagent ions and instrument operation conditions.

The chemical ionization gas entering the ion source is produced by mixing $NH_3$ and $H_2O$ from two streams: a 20 sccm

flow of water vapor from the headspace of a liquid water reservoir (denoted as $H_2O$ flow) and an additional 1 sccm from the headspace of a reservoir containing 0.5% (vol %) ammonium hydroxide water solution (denoted as $NH_3$ flow, which contains both $NH_3$ and $H_2O$). The ion source consists of two conical surfaces with a voltage gradient. A plasma is produced between the conical surfaces, which primarily ionizes water molecules producing $H_3O^+$. The discharge current is regulated at 2.0 mA. Because $NH_3$ has a larger proton affinity than $H_2O$, the proton transfer reaction (Eqn. 1) produces $NH_4^+$, which then readily

clusters with abundant $H_2O$ to produce the targeted reagent ion $NH_4^+ \cdot H_2O$ (Eqn. 2). Besides $H_2O$, $NH_4^+$ can also cluster with $NH_3$ to produce $NH_4^+ \cdot NH_3$ (Eqn. 3). The abundance of $H_2O$ in the ion source also leads to formation of $H_3O^+ \cdot (H_2O)_n$ cluster ions (Eqn. 4). Overall, several ions, $NH_4^+ \cdot X_n$ (ligand X = $NH_3$ and $H_2O$, n = 0,1,2) and $H_3O^+ \cdot (H_2O)_n$, are generated from the ion source and can potentially serve as reagent ions.

$$H_3O^+ + NH_3 \rightarrow NH_4^+ + H_2O \tag{1}$$

$$NH_4^+ + nH_2O \rightarrow NH_4^+ \cdot (H_2O)_n \tag{2}$$

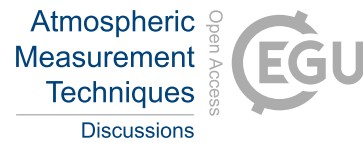

$$NH_4^+ + nNH_3 \rightarrow NH_4^+ \cdot (NH_3)_n \tag{3}$$

$$H_3O^+ + nH_2O \rightarrow H_3O^+ \cdot (H_2O)_n \tag{4}$$

The reagent gas flow pushes the ions into the FIMR where they subsequently react with analytes. Sample air enters the FIMR through a 10 mm long PEEK capillary (ID 0.18 mm). The sample flow rate is ∼100 sccm, at a FIMR pressure of 3 mbar in this study. The FIMR is a 100 mm long glass tube with an inner diameter of 10 mm. A quadrupole radio frequency (RF) field is applied to the FIMR to collimate ions into a narrow beam significantly enhancing the sensitivity (Krechmer et al., 2018). The FIMR conditions, including temperature, pressure, drift voltage, and the ratio of $NH_3$ to $H_2O$ into the ion source, all control the degree of cluster-ion formation, the distribution of reagent ions, and ultimately the sensitivity, as will be discussed in section 3.2 and 3.3. Ions from the FIMR travel through a big segmented quadrupole (BSQ). The BSQ serves as a high-pass band filter to reduce the signal intensity of reagent ions while simultaneously guiding ions into the time-of-flight mass spectrometer. As a result of this filtering, the observed distribution of reagent ions is not the same as the actual distribution in the FIMR (Krechmer et al., 2018). After the BSQ, the ions travel through the primary beam region and eventually are detected by the long time-of-flight mass spectrometer with a mass resolution (full width at half maximum, FWHM) up to 8000 at m/Q 100. The extraction frequency of the ToF is set at 17.5 kHz.

## 2.2 Laboratory Characterization

We calibrate the instrumental sensitivities (counts per second per ppbv, cps ppbv$^{-1}$) of 60 organic compounds (Table 1) using two methods, standard gas cylinders (SGC) and a home-built liquid calibration unit (LCU). The standard gas cylinders (Apel Riemer Environmental, Inc.) have a 5% analytical accuracy and 10% blend tolerance. The LCU is described in Coggon et al. (2018) and it utilizes a high precision syringe pump (Harvard Apparatus) to inject an aqueous solution with known concentrations of analytes at controlled flow rates (0-100 nL min$^{-1}$) into a heated zero air stream (0 - 5 L min$^{-1}$) at 60°C. The aqueous droplets evaporate and result in a gas flow containing analytes at defined concentration, which is sampled by the $NH_4^+$ CIMS. The aqueous standard mixture is prepared using either water (LCU-W) or hexane (LCU-H) as the solvent, depending on the solubility and the reactivity of analytes. 19 compounds are calibrated using both methods and show good agreement. We find minimal dependence of sensitivity on sample relative humidity (RH), consistent with the observations made when running the Vocus in $H_3O^+$ mode (Krechmer et al., 2018). This is mainly because a large amount of water vapor (20 sccm) is deliberately added to the FIMR. As an example, the water amount in 100 sccm ambient sample under 25°C and 100% RH is only 15% of the added 20 sccm water vapor to the FIMR, assuming no water vapor loss in both processes. The instrument background is determined by passing ambient air through a platinum catalytic converter heated to 400 °C. The detection limit is defined as three standard deviations of measurement background for 1 s integration time.



Table 1: Sensitivities (cps ppbv$^{-1}$), background (cps), and detection limits (pptv for a 1 s integration time) of NH$_4^+$ CIMS

| Species | Ion Formula | Ion m/Q | Sensitivity | Background | LOD | Methods |
|---|---|---|---|---|---|---|
| Methanol | NH$_4^+ \cdot$CH$_4$O | 50.06 | <1 | 5 | 2.5e4 | SGC, LCU-W |
| Acetonitrile | NH$_4^+ \cdot$C$_2$H$_3$N | 59.06 | 5.5e2 | 1.3e2 | 85 | SGC, LCU-W |
| Acetaldehyde | NH$_4^+ \cdot$C$_2$H$_4$O | 62.06 | 21 | 2.5e2 | 3.2e3 | SGC, LCU-H |
| Ethylene oxide | NH$_4^+ \cdot$C$_2$H$_4$O | 62.06 | <1 | 2.5e2 | 2.0e5 | SGC |
| Ethanol | NH$_4^+ \cdot$C$_2$H$_6$O | 64.08 | 7 | 1.5e2 | 6.9e3 | SGC, LCU-W |
| Propionitrile | NH$_4^+ \cdot$C$_3$H$_5$N | 73.11 | 1.8e3 | N/A | N/A | SGC |
| Acrolein | NH$_4^+ \cdot$C$_3$H$_4$O | 74.06 | 2.1e2 | 31 | 1.7e2 | SGC |
| Acetone | NH$_4^+ \cdot$C$_3$H$_6$O | 76.08 | 1.2e3 | 4.7e3 | 2.3e2 | SGC, LCU-W |
| Propanal | NH$_4^+ \cdot$C$_3$H$_6$O | 76.08 | 1.0e2 | 4.7e3 | 2.8e3 | SGC |
| Acetic Acid | NH$_4^+ \cdot$C$_2$H$_4$O$_2$ | 78.05 | 25 | 3.6e2 | 3.9e3 | LCU-W |
| 2-propanol | NH$_4^+ \cdot$C$_3$H$_8$O | 78.09 | 90 | 23 | 2.9e2 | LCU-W |
| Ethylene Glycol | NH$_4^+ \cdot$C$_2$H$_6$O$_2$ | 80.07 | 1.0e3 | 95 | 38 | LCU-W |
| Furan | NH$_4^+ \cdot$C$_4$H$_4$O | 86.06 | <1 | 13 | 4.4e4 | SGC, LCU-H |
| Isoprene | NH$_4^+ \cdot$C$_5$H$_8$ | 86.10 | 28 | 2 | 1.7e2 | SGC |
| MVK | NH$_4^+ \cdot$C$_4$H$_6$O | 88.08 | 1.5e3 | 40 | 18 | SGC, LCU-W |
| MACR | NH$_4^+ \cdot$C$_4$H$_6$O | 88.08 | 3.3e2 | 40 | 84 | SGC, LCU-H |
| MEK | NH$_4^+ \cdot$C$_4$H$_8$O | 90.14 | 1.6e3 | 92 | 22 | SGC |
| Tetrahydrofuran | NH$_4^+ \cdot$C$_4$H$_8$O | 90.14 | 8.2e2 | 92 | 44 | SGC |
| Propanoic Acid | NH$_4^+ \cdot$C$_3$H$_6$O$_2$ | 92.07 | 3.1e2 | 2.9e2 | 2.3e2 | LCU-W |
| Hydroxyacetone | NH$_4^+ \cdot$C$_3$H$_6$O$_2$ | 92.07 | 2.1e3 | 2.9e2 | 35 | SGC, LCU-W |
| 2-butanol | NH$_4^+ \cdot$C$_4$H$_{10}$O | 92.11 | 1.9e2 | 4 | 47 | LCU-W |
| 1,3-propanediol | NH$_4^+ \cdot$C$_3$H$_8$O$_2$ | 94.09 | 1.0e3 | 3.6e2 | 68 | LCU-W |
| Benzene | NH$_4^+ \cdot$C$_6$H$_6$ | 96.08 | <1 | 3 | 9.3e3 | SGC, LCU-H |
| 2-methylfuran | NH$_4^+ \cdot$C$_5$H$_6$O | 100.08 | 37 | 18 | 4.8e2 | SGC, LCU-H |
| Methacrylic Acid | NH$_4^+ \cdot$C$_4$H$_6$O$_2$ | 104.07 | 97 | 1.6e2 | 5.0e2 | LCU-W |
| Pentanal | NH$_4^+ \cdot$C$_5$H$_{10}$O | 104.11 | 2.2e2 | 22 | 90 | LCU-H |
| 3-Pentanone | NH$_4^+ \cdot$C$_5$H$_{10}$O | 104.11 | 2.9e3 | 22 | 7 | LCU-H |
| 2-Pentanone | NH$_4^+ \cdot$C$_5$H$_{10}$O | 104.11 | 2.8e3 | 22 | 7 | SGC |
| 2,3-butanedione | NH$_4^+ \cdot$C$_4$H$_6$O$_2$ | 104.12 | 2.6e2 | 1.6e2 | 1.8e2 | LCU-W |
| Butyric Acid | NH$_4^+ \cdot$C$_4$H$_8$O$_2$ | 106.09 | 1.8e2 | 74 | 2.0e2 | LCU-W |
| 2-pentanol | NH$_4^+ \cdot$C$_5$H$_{12}$O | 106.12 | 3.0e2 | 2 | 19 | LCU-W |





**Table 1 – continued from previous page**

| Species | Ion Formula | Ion m/Q | Sensitivity | Background | LOD | Methods |
|---|---|---|---|---|---|---|
| Toluene | $NH_4^+ \cdot C_7H_8$ | 110.10 | <1 | 2 | 1.3e4 | SGC, LCU-H |
| Phenol | $NH_4^+ \cdot C_6H_6O$ | 112.08 | 1.9e2 | 19 | 1.2e2 | SGC, LCU-H |
| Furfural | $NH_4^+ \cdot C_5H_4O_2$ | 114.06 | 3.3e3 | 15 | 5 | SGC |
| 2-hexanone | $NH_4^+ \cdot C_6H_{12}O$ | 118.12 | 3.8e3 | 10 | 4 | SGC, LCU-H |
| 2,3-Pentanedione | $NH_4^+ \cdot C_5H_8O_2$ | 118.15 | 4.9e2 | 80 | 76 | SGC, LCU-W |
| Hexanal | $NH_4^+ \cdot C_6H_{12}O$ | 118.19 | 7.5e2 | 10 | 18 | LCU-H |
| 1-hexanol | $NH_4^+ \cdot C_6H_{14}O$ | 120.14 | 1.4e2 | 1 | 36 | LCU-W |
| Benzonitrile | $NH_4^+ \cdot C_7H_5N$ | 121.08 | 3.7e3 | 2 | 3 | SGC, LCU-H |
| Styrene | $NH_4^+ \cdot C_8H_8$ | 122.10 | 4.2e2 | 4 | 29 | SGC |
| Benzaldehyde | $NH_4^+ \cdot C_7H_6O$ | 124.08 | 1.9e3 | 6 | 30 | SGC, LCU-H |
| o-xylene | $NH_4^+ \cdot C_8H_{10}$ | 124.11 | <1 | 2 | 4.5e4 | SGC |
| m-xylene | $NH_4^+ \cdot C_8H_{10}$ | 124.11 | <1 | 2 | 2.1e4 | SGC |
| 2-methylphenol | $NH_4^+ \cdot C_7H_8O$ | 126.09 | 2.5e2 | 5 | 48 | SGC |
| heptanal | $NH_4^+ \cdot C_7H_{14}O$ | 132.22 | 6.5e2 | 6 | 15 | LCU-H |
| 2-heptanone | $NH_4^+ \cdot C_7H_{14}O$ | 132.22 | 3.5e3 | 6 | 3 | LCU-H |
| 1,2,4-TMB | $NH_4^+ \cdot C_9H_{12}$ | 138.13 | <1 | 0.6 | 2.2e3 | SGC |
| Naphthalene | $NH_4^+ \cdot C_{10}H_8$ | 146.20 | 6 | 1.5 | 1.9e3 | SGC |
| Octanal | $NH_4^+ \cdot C_8H_{16}O$ | 146.24 | 8.0e2 | 5 | 11 | LCU-H |
| 2-octanone | $NH_4^+ \cdot C_8H_{16}O$ | 146.24 | 2.9e3 | 5 | 3 | LCU-H |
| p-cymeme | $NH_4^+ \cdot C_{10}H_{14}$ | 152.25 | 9 | 0.8 | 4.0e2 | SGC |
| Limonene | $NH_4^+ \cdot C_{10}H_{16}$ | 154.16 | 3.9e2 | 2 | 11 | SGC, LCU-H |
| α-pinene | $NH_4^+ \cdot C_{10}H_{16}$ | 154.16 | 3.6e2 | 2 | 12 | LCU-H |
| β-pinene | $NH_4^+ \cdot C_{10}H_{16}$ | 154.16 | 4.6e2 | 2 | 9 | SGC |
| Camphene | $NH_4^+ \cdot C_{10}H_{16}$ | 154.16 | 3.4e2 | 2 | 13 | SGC |
| Nonanal | $NH_4^+ \cdot C_9H_{18}O$ | 160.27 | 6.5e2 | 7 | 18 | LCU-H |
| 2-nonanone | $NH_4^+ \cdot C_9H_{18}O$ | 160.27 | 2.6e3 | 7 | 4 | LCU-H |
| α-pinene oxide | $NH_4^+ \cdot C_{10}H_{16}O$ | 170.27 | 1.1e3 | 2 | 4 | LCU-H |
| Texanol | $NH_4^+ \cdot C_{12}H_{24}O_3$ | 234.21 | 9.0e2 | 2 | 6 | LCU-W |
| D5-siloxane | $NH_4^+ \cdot C_{10}H_{30}O_5Si_5$ | 388.81 | 6.2e3 | 5 | 1 | SGC, LCU-H |

During transport, ions get lost in the BSQ, in the ion guides, and in the extraction region of the ToF. We quantify the mass-dependent transmission efficiency relative to the reagent ion $NH_4^+ \cdot H_2O$ by introducing a series of compounds spanning a range





of molecular weight (32 - 370 m/Q) in a large enough quantity to deplete the fraction of reagent ions by ∼20-30% (Huey et al., 1995; Heinritzi et al., 2016). The ratio of the increase of the product ions to the decrease of the reagent ion indicates the relative transmission efficiency between these two masses. A detailed derivation can be found in the Supplement S1.

To probe the stability of product ions, we performed voltage scanning tests following the procedure outlined in Lopez-Hilfiker et al. (2016) and Zaytsev et al. (2019). In brief, we vary the voltage gradient ($\Delta V$) between FIMR back and skimmer while keeping the voltage gradient between FIMR front and back constant. A larger $\Delta V$ increases the collisional energy, causes stronger collision-induced dissociation of product ions, and tends to decrease the the signal of product ions. We define $\Delta V_{50}$ as the voltage gradient at which the product ion signal drops to half of the maximum signal. Following the procedures in Zaytsev

et al. (2019) and outlined in Supplement S3, $\Delta V_{50}$ is converted to the kinetic energy of product ions in the center of mass ($KE_{cm,50}$), which is a measure of their stability.

### 2.3    Field Deployment

The $NH_4^+$ CIMS was deployed during the RECAP campaign (Re-Evaluating the Chemistry of Air Pollutants in California) in Pasadena, California from August-September, 2021. The ground sampling site is located on the campus of the California

Institute of Technology, which is only one block away from the original sampling site during the 2010 Calnex study (Ryerson et al., 2013). The instrument inlet was set up on a tower 10 m above the ground. The instrument was operated to sample gas phase from August 10th to 19th. Later, the instrument was coupled to a Vocus Inlet for Aerosol (VIA) to automatically switch sampling between gas and particle phases. This study will focus on the gas phase sampling period. Co-located instruments of relevance to this study include a Proton-Transfer-Reaction Mass Spectrometry (PTR-MS) (Yuan et al., 2016; de Gouw and

Warneke, 2007), an iodide Chemical Ionization Mass Spectrometry ($I^-$ CIMS) (Veres et al., 2020; Robinson et al., 2022), and a Gas-Chromatography Mass Spectrometry (GC-MS) (Lerner et al., 2017). The PTR-MS replaced the traditional drift tube with the same FIMR as used in the $NH_4^+$ CIMS.

### 3    Instrument Performance

### 3.1    Overview of Ion Chemistry

The target primary reagent ion is $NH_4^+ \cdot H_2O$, which ionizes analyes (A) primarily via ligand-switching reactions (Eqn. 5) to form product ion $NH_4^+ \cdot A$. As analogous to proton affinity, we define $NH_4^+$ affinity as the negative of the enthalpy change in the reaction between $NH_4^+$ and an analyte. If an analyte has a larger $NH_4^+$ affinity than $H_2O$, reaction (5) is exothermic and will occur at a rate close to the collision limit when the difference in $NH_4^+$ affinity is sufficiently large (Adams et al., 2003). Otherwise, the ligand-switching reaction is endothermic. The energy imparted via the drift voltage could aid the endothermic

reaction to overcome the energy barrier, but the instrument sensitivity in these instances is expected to be low. The desired product ion is a cluster with ammonium (i.e., $NH_4^+ \cdot A$). Due to the presence of electric fields, $NH_4^+ \cdot A$ may fragment via energetic collisions. This process affects the product distribution and the instrument sensitivity. Besides the target primary ion





$NH_4^+ \cdot H_2O$, ions $NH_4^+ \cdot X_n$ (X = $NH_3$ and $H_2O$) and $H_3O^+ \cdot (H_2O)_n$ (n = 0,1,2) are observed, because the chemical ionization gas supply is a mixture of $NH_3$ and $H_2O$. These ions can also serve as reagent ions. Compared to $NH_4^+$, $NH_4^+ \cdot H_2O$ ionization

is softer, because the $H_2O$ acts as a third-body which dissipates some reaction energy. The reactivities of $NH_4^+ \cdot H_2O$ and $NH_4^+ \cdot NH_3$ are also expected to be different, as the $NH_3$ has a larger $NH_4^+$ affinity than $H_2O$ does (i.e., 108 vs 86 kJ mol[-1], NIST Chemistry WebBook). Therefore, the presence of multiple reagent ions will complicate the ionization chemistry and the interpretation of the mass spectra. To avoid such complication, the instrument conditions need to be carefully optimized to ensure $NH_4^+ \cdot H_2O$ exists as the dominant ion reacting with analytes.

$$NH_4^+ \cdot H_2O + A \rightarrow NH_4^+ \cdot A + H_2O \tag{5}$$

### 3.2 Modeling the Distribution of Reagent Ions

The distribution of the reagent ions is controlled by several factors, including the FIMR reduced electric field (E/N), temperature (T), pressure (P), the $H_2O$ mixing ratio ($\chi_{H_2O}$), and the ratio of $NH_3$ to $H_2O$ ($NH_3/H_2O$). Many of these factors are interdependent - e.g., the E/N depends on pressure and temperature. To unravel the influences of these factors on the distribu-

tion of reagent ions, we develop a kinetic model. The model includes a series of reactions between two ions ($NH_4^+$ and $H_3O^+$) and two neutral molecules ($NH_3$ and $H_2O$). Clusters containing up to three molecules are considered, which leads to a total of 14 different ion clusters (Figure S3). The ion-molecule cluster reaction rate constant (i.e., forward reaction with $k_{forward}$) is calculated using the parameterization in Su (1994), assuming the reaction proceeds at the collision limit. The reaction rate constant of the declustering reaction (i.e., reverse reaction with $k_{reverse}$) is calculated using $k_{forward}$ and the equilibrium con-

stant $K_{eq}$. $k_{reverse}$ for reaction 6, for example, is expressed by Eqn. 7, where $M_0$ represents the number density (cm[-3]) under standard condition and $K_{eq}$ represents the reaction equilibrium constant. $K_{eq}$ is calculated using Eqn.8, where $\Delta H^0$ and $\Delta S^0$ represent the enthalpy and entropy changes of the reaction at standard condition, respectively (Table S1), and $T_{eff}$ represents the effective temperature of the ions in the FIMR. $T_{eff}$ is calculated using Eqn. 9 (de Gouw et al., 2003), where $k_B$ is the Boltzmann constant, $m_{I^+}$, $m_A$, and $m_{buffer}$ are the masses of the ion $I^+$, the neutral analyte A, and the buffer gas, respectively,

and the $\nu_d$ is the drift velocity of ion $IA^+$. $\nu_d$ is calculated using Eqn. 10, where $\mu_0$ is the reduced mobility of $IA^+$ and calculated based on the parameterization in Steiner et al. (2014), P and T are the FIMR pressure and temperature, respectively, and E is the electric field strength across the FIMR.

$$I^+ + A \underset{k_{reverse}}{\overset{k_{forward}}{\rightleftharpoons}} IA^+ \tag{6}$$

$$k_{reverse} = \frac{k_{forward} \times M_0}{K_{eq}} \tag{7}$$

$$K_{eq} = \exp(-\frac{\Delta H^0}{RT_{eff}} + \frac{\Delta S^0}{R}) \tag{8}$$





$$\frac{3}{2}k_B T_{eff} = \frac{3}{2}k_B T_{FIMR} + \frac{(m_{I^+} + m_{buffer})m_A}{(m_{I^+} + m_A)}\frac{v_d^2}{2} \tag{9}$$

$$v_d = \mu_0 \frac{P_0}{P}\frac{T}{T_0}E \tag{10}$$

The influences of different FIMR conditions (i.e., E/N, T, P, $\chi_{H_2O}$, and $NH_3/H_2O$) on the distribution of reagent ions are intertwined. To visualize their impacts, we first conduct simulations covering wide ranges of all five factors to locate the

condition yielding the largest fraction of $NH_4^+ \cdot H_2O$ in total ions (denoted as $f_{NH_4^+ \cdot H_2O}$). The optimized condition is E/N = 60 Td (Townsend), T = 330 K, P = 5 mbar, $\chi_{H_2O}$ = 0.25, and $NH_3/H_2O$ = 0.1%. Then, we conduct simulations using the optimal condition as a start point and vary one factor at a time while holding the other four constant, to investigate the impact of each factor on the distribution.

The simulation results are shown in Figure 1. Figure 1a shows that the reduced electric field (E/N) strongly impacts the

distribution of reagent ions. When the E/N is below 40 Td, $H_3O^+ \cdot (H_2O)_3$ is the dominant ion, because the electric field is too weak to decluster. When the E/N is above 80 Td, $NH_4^+$ is dominant, because the electric field results in strong declustering and because $NH_3$ has higher proton affinity than $H_2O$. Only within a narrow E/N window (50 - 65 Td) is the target reagent ion $NH_4^+ \cdot H_2O$ the most abundant ion. Within this window, several other ions also exist, including $NH_4^+$, $NH_4^+ \cdot NH_3$, and $NH_4^+ \cdot (H_2O)_2$. The FIMR P and T impact the distribution (Figure 1b and c) through a similar mechanism as E/N, as smaller P

and larger T results in larger E/N. As a result, $f_{NH_4^+ \cdot H_2O}$ also exhibits a non-monotonic dependence on the FIMR P and T. The impact of $H_2O$ mixing ratio in the FIMR ($\chi_{H_2O}$) on the distribution is shown in Figure 1d. The $f_{NH_4^+ \cdot H_2O}$ initially increases with the $\chi_{H_2O}$, reaches a maximum when $\chi_{H_2O}$ is roughly 0.16-0.18, and then decreases with increasing $\chi_{H_2O}$. This trend is because low $\chi_{H_2O}$ limits the supply of $H_2O$ to cluster with $NH_4^+$ and high $\chi_{H_2O}$ favors the formation of larger clusters. To illustrate, Figure 1d shows that as $\chi_{H_2O}$ increases, the fraction of smaller clusters (i.e., $NH_4^+ \cdot H_2O$) decreases, but the fraction of larger

clusters (i.e., $NH_4^+ \cdot (H_2O)_2$ and $NH_4^+ \cdot (H_2O)_3$) increases. Lastly, the $NH_3/H_2O$ ratio has a strong impact on the cluster ion distribution (Figure 1e). Low $NH_3/H_2O$ ratio (< 0.2%) results in insufficient supply of $NH_4^+$ and therefore $H_3O^+ \cdot (H_2O)_n$ ions dominate. High $NH_3/H_2O$ ratio (> 0.55%) causes $NH_4^+$ to mainly cluster with $NH_3$, producing large amounts of $NH_4^+ \cdot NH_3$.

Evaluation of the kinetic simulation results by experimental observations is desirable, but challenging. One challenge is that the distribution of reagent ions can not be measured, because the BSQ serves as a high-pass band filter which reduces

the signal intensity of reagent ions. Another challenge is that voltages in the ion transfer region between the drift tube and the mass analyzer can change the distribution of reagent ions, which causes the measured distribution different from that in the FIMR (Krechmer et al., 2018; Breitenlechner et al., 2022; Yuan et al., 2016). Overall, the simulation results illustrate the controlling effects of FIMR conditions on the distribution of reagent ions. The determination of FIMR conditions is eventually based on experimental calibration of instrumental sensitivity, which can be guided by the modeled distribution of reagent ions,

as discussed in next section.

**Figure 1.** The dependence of modeled distribution of reagent ions on FIMR conditions. (a) E/N; (b) P; (c) T; (d) $H_2O$ mixing ratio; (e) $NH_3/H_2O$. In each panel, the other four factors are held constant at the following conditions: E/N = 60 Td, P = 5 mbar, T = 330 K, $H_2O$ mixing ratio = 0.25, $NH_3/H_2O$ = 0.1%. Because the impacts of these factors are intertwined, each panel will change if the other four factors are at different values, as an example shown in Figure S4.

## 3.3 Dependence of sensitivities on FIMR conditions

While the above section modeled the dependence of the distribution of reagent ions on FIMR conditions, in this section we experimentally evaluate the dependence of analyte sensitivities on FIMR conditions, including E/N, pressure, temperature, and $NH_3/H_2O$ ratio. The analyte sensitivity depends not only on the distribution of reagent ions, but also other factors, including the number density of analytes in the FIMR, ion-molecular reaction time, stability of the product ion, and the transmission





efficiency of product ions, as discussed below. Similar to the analysis in kinetic modeling, we experimentally vary one factor while holding the others constant.

Figure 2a shows the impacts of E/N on sensitivities of representative analytes. The E/N is varied by ramping the FIMR front voltage from 100 to 600 V, while holding the FIMR back voltage at 5 V. Under a FIMR pressure and temperature of
3 mbar and 314 K, respectively, the E/N ranges from 13 to 83 Td. The dependence of sensitivities on E/N follows a similar trend of the modeled distribution of $NH_4^+ \cdot H_2O$ (Figure 1a). The sensitivities initially increase with increasing E/N, partly because of more reagent ion $NH_4^+ \cdot H_2O$. As E/N keeps increasing, $NH_4^+ \cdot H_2O$ declusters into $NH_4^+$, so less $NH_4^+ \cdot H_2O$ causes a decrease in sensitivities. Besides changing the distribution of the reagent ions, changing E/N influences the sensitivity via other mechanisms. The E/N influences the focusing effect of ions in the FIMR. Krechmer et al. (2018) shows that the higher E/N
better focuses ions to the central axis of the reactor and increases the sensitivity. This may explain the uptick in sensitivities when E/N increases from 80 to 90 Td, which is not observed in the modeled $NH_4^+ \cdot H_2O$. In addition, E/N affects the extent of declustering of $NH_4^+ \cdot A$ in the FIMR. Overall, the observed dependence of sensitivities on E/N is a superposition of at least three effects, including distribution of reagent ions, focusing effects, and the extent of declustering.

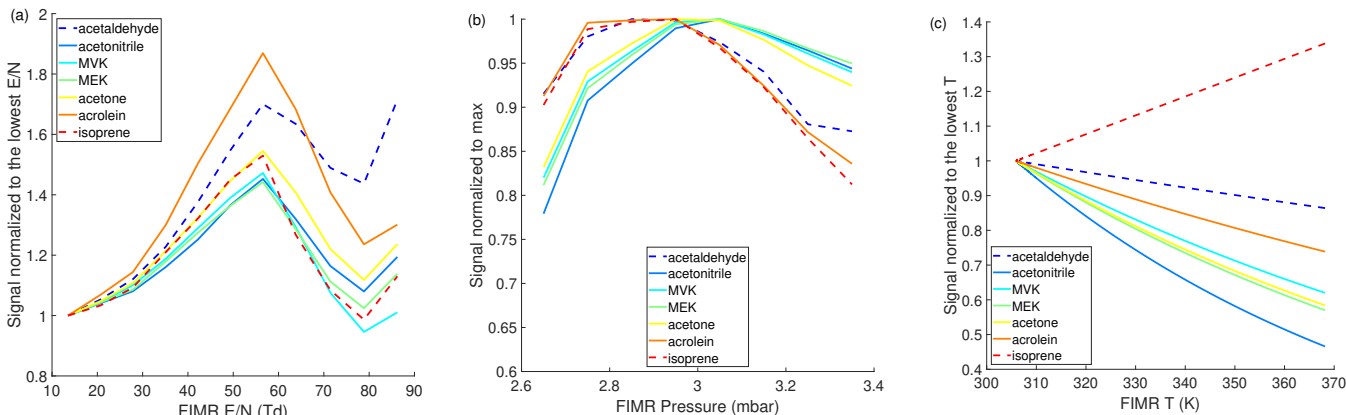

**Figure 2.** Dependence of instrument sensitivities of representative species on FIMR conditions (a) E/N; (b) P; (c) T. The range of E/N in panel (a) is obtained by varying the drift voltage while maintaining the P and T at 3 mbar and 313 K, respectively. Analytes with sensitivities lower than 50 cps ppbv$^{-1}$ are shown in dashed lines. The parent ion $NH_4^+ \cdot A$ is used to quantify the sensitivity.

The effects of FIMR pressure on sensitivities are shown in Figure 2b. The sensitivities exhibit a non-monotonic dependence
on FIMR pressure, in a similar manner as the reagent ion $NH_4^+ \cdot H_2O$ does (Figure 1b), suggesting the pressure-dependent sensitivities are related to the pressure-dependent distribution of reagent ions. In addition, higher pressure increases the number density of analyte molecules in the FIMR, which tends to increase the sensitivity. However, this effect is smaller than the effect of changing reagent ion on sensitivities, as Figure 2b shows that the sensitivities decrease with increasing pressure beyond 3 mbar.

The effects of FIMR temperature on sensitivities are shown in Figure 2c. Among the seven compounds tested here, the sensitivities of six oxygenated compounds exhibit a negative dependence on the temperature between 310 and 370 K. The re-





duced VOC, isoprene, exhibits a positive dependence. Similar to isoprene, α-pinene sensitivity also increases with temperature in the 303 - 350 K window as recently reported in Khare et al. (2022). Here we examine the opposite trends of temperature-dependent sensitivity between acetone and α-pinene, because their $NH_4^+$ affinities are available in the literature (Supplement S4). α-pinene has a $NH_4^+$ affinity smaller than that of $H_2O$ (i.e., 75 vs 86 kJ mol$^{-1}$ from Canaval et al. (2019)), resulting in the ligand-switching reaction between α-pinene and $NH_4^+ \cdot H_2O$ being endothermic. Therefore, the reaction is promoted under higher temperature, which enhances the sensitivity. In contrast, the ligand-switching reaction between acetone and $NH_4^+ \cdot H_2O$ is exothermic, because acetone has a larger $NH_4^+$ affinity than $H_2O$ (i.e., 110 vs 86 kJ mol$^{-1}$ from Canaval et al. (2019)). For exothermic reactions (ΔH is negative), higher temperature leads to smaller $K_{eq}$ (Eqn. 8), larger $k_{reverse}$ (Eqn. 7), and hence lower sensitivity. To better understand the temperature-dependent sensitivities, we add the reversible reactions of acetone and α-pinene with $NH_4^+ \cdot H_2O$ to the kinetic model depicted in Figure S3 and simulate the dependence of their sensitivities on temperature. As shown in Figure S5, the model can reproduce the observed dependence of their sensitivities on temperature. The $NH_4^+$ affinity of isoprene is not available, but it is expected to be even smaller than α-pinene, given that the isoprene sensitivity is 10 times smaller than that of α-pinene. Thus, the reaction between isoprene and $NH_4^+ \cdot H_2O$ is likely also endothermic, causing the increasing sensitivity with higher temperature as shown in Figure 2c.

The effects of $NH_3/H_2O$ ratio on sensitivities are experimentally tested by simultaneously varying the flow rates of $NH_3$ and $H_2O$, while keeping the total flow rate constant. Because the $NH_3$ flow is a mixture of $NH_3$ and $H_2O$, the accurate flow rate of $NH_3$ is unknown. We use the observed ratio of $NH_4^+ \cdot H_2O/H_3O^+ \cdot H_2O$ and $NH_4^+ \cdot NH_3/NH_4^+ \cdot H_2O$ to approximate the $NH_3/H_2O$ ratio, because these three ions have similar transmission efficiency and their relative abundance directly depends on the $NH_3/H_2O$ ratio. Figure 3 shows the dependence of sensitivities of nearly 50 analytes on the $NH_4^+ \cdot H_2O/H_3O^+ \cdot H_2O$. For the majority of compounds, their sensitivities initially increase with $NH_4^+ \cdot H_2O/H_3O^+ \cdot H_2O$ and then show a decreasing trend. This trend is caused by the fact that the initial increase in $NH_3/H_2O$ favors the formation of $NH_4^+ \cdot H_2O$ and hence higher sensitivity, but high $NH_3/H_2O$ produces more $NH_4^+ \cdot NH_3$ clusters, leading to reduced sensitivity (Figure 1e). Taking acetone as an example, its $NH_4^+$ affinity (110 kJ mol$^{-1}$) is higher than that of $H_2O$ (86 kJ mol$^{-1}$), but close to that of $NH_3$ (108 kJ mol$^{-1}$). As a result, the ligand-switching reaction between acetone and $NH_4^+ \cdot NH_3$ is less favorable than that between acetone and $NH_4^+ \cdot H_2O$. The sensitivities of several compounds, including D5-siloxane, texanol, and several monoterpenes, exhibit a monotonic increase with $NH_4^+ \cdot H_2O/H_3O^+ \cdot H_2O$ ratio within the tested range, but will likely decrease at a higher $NH_4^+ \cdot H_2O/H_3O^+ \cdot H_2O$ ratio. The maximum sensitivity occurs at different $NH_4^+ \cdot H_2O/H_3O^+ \cdot H_2O$ ratios for different compounds, likely because they have different reactivities towards $NH_4^+ \cdot H_2O$ and other reagent ions.

Unlike the other four factors (i.e., E/N, T, P, and $\chi_{H_2O}$) which can be accurately controlled, the $NH_3/H_2O$ ratio and the resultant $NH_4^+ \cdot H_2O/H_3O^+ \cdot H_2O$ ratio change over time owing to the aging effects within the solution that supplies $NH_3$. In the current approach to supply the chemical ionization gas, the $NH_3/H_2O$ ratio is controlled by the combination of the concentration of ammonium hydroxide aqueous solution and flow rates from the water and ammonium hydroxide reservoirs. Because $NH_3$ is more volatile than $H_2O$, the concentration of the ammonium hydroxide water solution decrease over time, resulting in a decreasing trend of $NH_3/H_2O$ over timescale of weeks. In addition, the temperature variation of the ammonium hydroxide water solution changes the partitioning of $NH_3$ and hence the $NH_3/H_2O$ ratio. One approach to compensate for the $NH_3$ loss is



to adjust the flow rate from the ammonium hydroxide reservoir to maintain a relatively constant $NH_4^+ \cdot H_2O/H_3O^+ \cdot H_2O$ ratio. As shown in Figure 3, for most compounds studied here, the largest sensitivity occurs when the $NH_4^+ \cdot H_2O/H_3O^+ \cdot H_2O$ is between 5 and 20 and the sensitivity remains relatively constant in this window. Thus, we suggest to operate the instrument within this

$NH_4^+ \cdot H_2O/H_3O^+ \cdot H_2O$ range. Moreover, the instrument sensitivities should be calibrated as a function of $NH_4^+ \cdot H_2O/H_3O^+ \cdot H_2O$ ratio. Future studies exploring approaches to reliably supply chemical ionization gas are warranted.

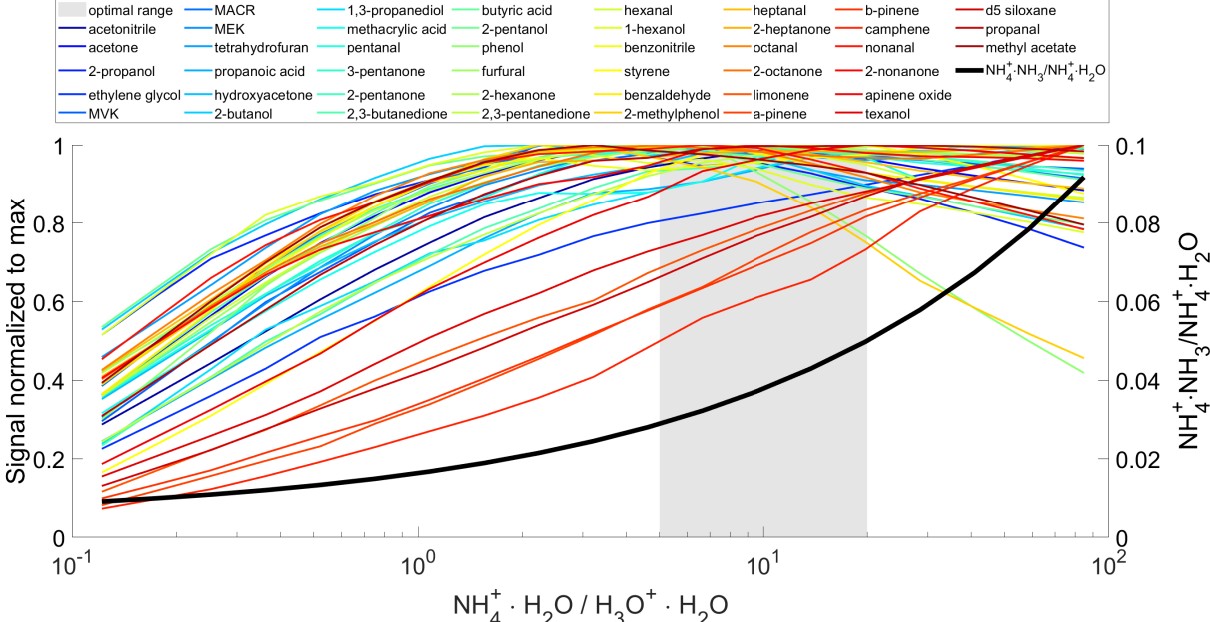

**Figure 3.** The effects of reagent ion distribution on sensitivities of various organic species. The sensitivity of each analyte is normalized to the corresponding maximum value. Only analytes with sensitivity larger than 50 cps ppbv$^{-1}$ are shown here. The grey area highlights the recommended $NH_4^+ \cdot H_2O/H_3O^+ \cdot H_2O$ range.

The impacts of various FIMR conditions on instrument sensitivities are highly intertwined. The relationship between instrument sensitivity and individual FIMR condition shown in Figure 2 could change when other FIMR conditions change. The optimal FIMR conditions should be explored collectively and systematically. The optimal condition for our instrument is

FIMR drift voltage 55 Td, 3 mbar, 40°C, 1 sccm from 0.5% ammonium hydroxide aqueous solution, and 20 sccm water vapor. A temperature value that is slightly higher than ambient temperature is chosen for control purpose.

### 3.4   Product Distributions from the Ion-Molecule Reactions

The desired reagent ion is $NH_4^+ \cdot H_2O$ and the desired ion-molecule reaction is the ligand-switching reaction between $NH_4^+ \cdot H_2O$ and analyte A, which produces cluster $NH_4^+ \cdot A$ as the parent ion (Eqn. 5). However, the presence of several reagent ions in the

FIMR and the declustering of $NH_4^+ \cdot A$ in the electric field induce a variety of reactions and causes complex product distributions.



Besides the target parent ion $NH_4^+ \cdot A$, we observe the protonated product ($H^+ \cdot A$), analyte clusters ($NH_4^+ \cdot H_2O \cdot A$, $H_3O^+ \cdot A$, and $H_3O^+ \cdot H_2O \cdot A$), and fragmentation products. The potential ion-molecule reactions and product ions can be generally expressed by reactions 11 and 12.

$$
\begin{aligned}
NH_4^+ \cdot (NH_3)_x \cdot (H_2O)_y + A &\rightarrow NH_4^+ \cdot A \ + \ xNH_3 \ + \ yH_2O \\
&\rightarrow NH_4^+ \cdot H_2O \cdot A \ + \ xNH_3 \ + \ (y-1)H_2O \\
&\rightarrow H^+ \cdot A \ + \ (x+1)NH_3 \ + \ yH_2O \\
&\rightarrow H_3O^+ \cdot A \ + \ (x+1)NH_3 \ + \ (y-1)H_2O \\
&\rightarrow H_3O^+ \cdot H_2O \cdot A \ + \ (x+1)NH_3 \ + \ (y-2)H_2O \\
&\rightarrow fragments
\end{aligned} \tag{11}
$$

$$
\begin{aligned}
H_3O^+ \cdot (H_2O)_n + A &\rightarrow H^+ \cdot (H_2O)_m \cdot A \ + \ (n-m+1)H_2O \\
&\rightarrow fragments
\end{aligned} \tag{12}
$$


We perform laboratory tests and measure the product distribution of 60 organic compounds. The product ions are identified by sampling the headspace of a small vial containing pure analyte. A distance is kept between the instrument inlet and the vial to keep analyte concentration low. Ions correlating with the parent ion ($NH_4^+ \cdot A$) with $r^2$ larger than 0.95 and accounting for larger than 1% of the parent ion signal are considered as product ions from the analyte. The distribution of product ions depends on the distribution of reagent ions. In this test, we maintain the $NH_4^+ \cdot H_2O/H_3O^+ \cdot H_2O$ ratio between 5 and 20. Under this condition, the ion chemistry of $H_3O^+ \cdot (H_2O)_m$ is negligible (Eqn. 12).

Figure 4 shows the product distributions for all tested analytes, grouped by their chemical class. The analyte sensitivities are represented by the circle size in the figure. Among all classes, acids, ketones, and nitriles have the most desirable product distribution, in which the fraction of parent ion $NH_4^+ \cdot A$ in all product ions (denoted as $f_{NH_4^+ \cdot A}$) is more than 90%, with the exceptions of acetic acid. For 2-octanone and 2-nonanone, $NH_4^+ \cdot A$ is the sole product ion. For the alcohols, the product distribution is diverse. 2-propanol and 2-butanol have fragmentation products ($NH_4^+ \cdot A - 2H$), which account for ∼5% of the total products, but the fragmentation mechanism is unclear. For the aldehydes, the $NH_4^+ \cdot A$ generally accounts for more than 80% of total product ions. The fraction tends to increase with larger molecules, for example, when comparing a homologous series of aldehydes (pentanal, hexanal, heptanal, octanal, and nonanal). Four monoterpenes studied here produce significant amount of protonated product ($H^+ \cdot A$), which is comparable to that of $NH_4^+ \cdot A$. Limonene produces ∼20% of fragmentation products. The causes of the product distributions of four monoterpenes are possibly explained by their proton affinity and $NH_4^+$ affinity. Three monoterpenes including α-pinene, β-pinene, and camphene have smaller $NH_4^+$ affinities than $H_2O$ (Table S2). Thus, their ligand-switching reactions with $NH_4^+ \cdot H_2O$ are endothermic and the production of $NH_4^+ \cdot A$ is likely aided by the energetic collision energy imparted by the drift voltage. These three monoterpenes have higher proton affinity than $NH_3$ (Table S2), so that $NH_4^+ \cdot A$ can undergo internal proton transfer to produce $AH^+ \cdot NH_3$, which breaks in the electric field and produces

$AH^+$. In contrast to the above three monoterpenes, limonene has larger $NH_4^+$ affinity than $H_2O$ and smaller proton affinity than $NH_3$ (Table S2). Thus, the ligand-switching reaction with $NH_4^+ \cdot H_2O$ is exothermic and the proton transfer reaction is thermodynamically unfavorable. The $H^+ \cdot$ limonene is likely produced from the declustering of $NH_4^+ \cdot$ limonene in the electric fields. The energy released from the exothermic reaction together with that imparted via the drift voltage could even break

$NH_4^+ \cdot$ limonene into fragments $C_6H_9^+$, $C_7H_{11}^+$, and $C_6H_{12}N^+$. For reduced aromatics (toluene, o-xylene, m-xylene, 1,2,4-TMB, and p-cymene), $H^+ \cdot A$ is the dominant product and $NH_4^+ \cdot A$ is negligible. The product distributions of reduced aromatics are puzzling, because these analytes have lower proton affinity than $NH_3$. Since their sensitivities are < 2 cps ppb$^{-1}$, it is not recommended to use $NH_4^+ \cdot H_2O$ to quantify reduced aromatics. Compared to reduced aromatics, oxygenated aromatics have higher sensitivity and larger $f_{NH_4^+ \cdot A}$. For example, benzaldehyde, 2-methylphenol, and furfural have $f_{NH_4^+ \cdot A}$ greater than 90%.

For a number of analytes in this study, the production of $NH_4^+ \cdot H_2O \cdot A$ is evident. This product complicates the interpretation of the mass spectra and introduces uncertainties in quantification, because the same ion is produced from an analyte (A) clustering with $NH_4^+ \cdot H_2O$ and an analyte with chemical formula $A+H_2O$ clustering with $NH_4^+$. For example, the ion $C_3H_{12}NO_2^+$ can be produced from either acetone ($C_3H_6O$) clustering with $NH_4^+ \cdot H_2O$ or 1,3-propanediol ($C_3H_8O_2$) clustering with $NH_4^+$. Cluster ions with $NH_4^+ \cdot NH_3$ are not observed for any compound. Overall, the product distribution is complicated and caution

is required in quantification.

### 3.5   Constraining the Sensitivity

Because of a lack of calibration standards, the $NH_4^+$ CIMS sensitivities towards the majority of routinely detected multifunctional organic compounds in the atmosphere are not quantifiable. We attempt to constrain the sensitivity building upon the extensive calibration of organic compounds from various chemical classes in this study. The observed instrument sensitivity

(S, counts per second per ppbv, cps ppbv$^{-1}$) is defined as the detected analyte signal (i.e., $[NH_4^+ \cdot A]$, cps) at a volume mixing ratio of 1 ppbv (parts per billion per volume). Fundamentally, S depends on the product ion formation and the transmission efficiency of product ions, as expressed by Eqn. 13 (Lopez-Hilfiker et al., 2016), where the integral represents the formation of product ions via the ion-molecule reactions in the IMR, $f_{NH_4^+ \cdot A}$ represents the fraction of parent ion $NH_4^+ \cdot A$ in all product ions, and TE represents the transmission efficiency of parent ion to the detector, which is dependent on the mass-to-charge ($\frac{m}{Q}$) and

the binding energy (B) of parent ion. In the integral, $[NH_4^+ \cdot H_2O]$ represents the $NH_4^+ \cdot H_2O$ concentration in the IMR, k and t represent the reaction rate constant and reaction time between reagent ion $NH_4^+ \cdot H_2O$ and analyte (A) in the IMR, respectively. Using this integral to represent the product ion formation is only valid when the ion-molecule reaction is in the kinetic-limited regime. In the thermodynamic regime, both forward and reverse ion-molecule reactions need to be considered.

$$S = \left( f_{NH_4^+ \cdot A} \times \int_0^t k \times [NH_4^+ \cdot H_2O] \, dt \right) \times \left( TE(\frac{m}{Q}, B) \right) \tag{13}$$

= parent ion formation $\times$ transmission efficiency





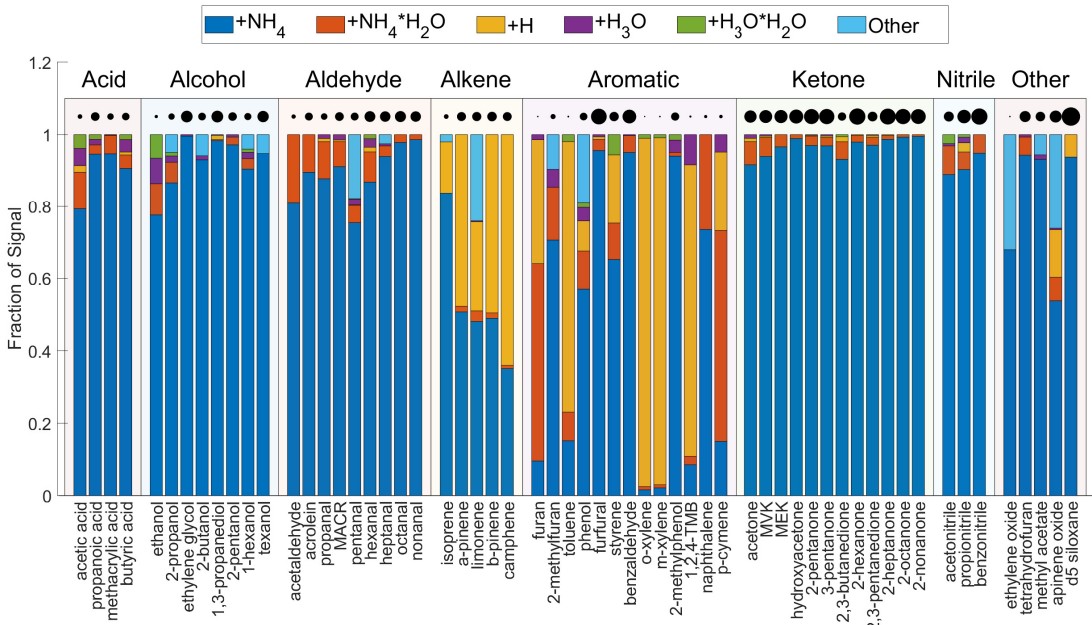

**Figure 4.** The product distributions of analytes in the $NH_4^+$ CIMS. The analytes are grouped by chemical class. Within each class, the analytes are sorted by increasing molecular weight. The distributions are obtained under the condition that the ratio of $NH_4^+ \cdot H_2O$ to $H_3O^+ \cdot H_2O$ is between 5 and 20. The product ion labeled "other" includes charge transfer products (e.g., $C_6H_6O^+$ for phenol), and fragmentation products (e.g., $C_5H_{12}N^+$ for pentanal). The product distribution of benzene is not shown because the signals of its product ions are too low to be reliably fitted. The circles are scaled to the square root of the analyte sensitivity.

$$TE(\frac{m}{Q}, B) = TE_{\frac{m}{Q}} \times TE_B \quad (14)$$

$$f(KE_{cm,50}) = TE_B = \frac{1}{\int_0^t [NH_4^+ \cdot H_2O] \, dt} \times \frac{S}{f_{NH_4^+ \cdot A} \times k \times TE_{\frac{m}{Q}}}$$
$$= \frac{1}{C} S_{corr} \quad (15)$$

Under a constant instrumental condition, the $[NH_4^+ \cdot H_2O]$ and reaction time are fixed. The sensitivity of an analyte is determined by $f_{NH_4^+ \cdot A}$, k, and TE. Among these three factors, $f_{NH_4^+ \cdot A}$ and k have less uncertain than TE. k for exothermic ligand-switching reactions is close to the collisional limit (Adams et al., 2003), which can be calculated according to Su (1994) using

the dipole moment and polarizability of the analyte (Table S3). $f_{NH_4^+ \cdot A}$ can be experimentally measured and it is close to 1 for multifunctional organic compounds, as discussed in Section 3.4. The TE, which represents the survival chance of ions through ion optics, is difficult to quantify. We assume the overall TE is represented by the product of $\frac{m}{Q}$-dependent TE (denoted as





TS$_\frac{m}{Q}$) and binding energy-dependent TE (denoted as TE$_B$) (Eqn. 14). TS$_\frac{m}{Q}$ represents the transmission efficiency through BSQ, extraction region of the ToF, and other processes that are dependent on $\frac{m}{Q}$. TS$_\frac{m}{Q}$ is experimentally quantified as described in

Section 2.2. TE$_B$ accounts for the ion loss via collision-induced dissociation caused by energy imparted by electric fields. TE$_B$ depends on the binding energy of the parent ion, as the parent ion with stronger bonds between analyte and NH$_4^+$ have a larger chance to survive the electric fields and hence a larger TE$_B$. Previous studies have revealed that the binding energies can be experimentally probed from the voltage scanning test (Lopez-Hilfiker et al., 2016; Zaytsev et al., 2019). In brief, the voltage gradient between two nearby ion optics (i.e., FIMR back and skimmer in this study) is systematically varied to obtain

the voltage gradient when the parent ion (NH$_4^+ \cdot$ A) signal drops by half (denoted as $\Delta V_{50}$). This $\Delta V_{50}$ represents the electric field required to break each NH$_4^+ \cdot$ A and therefore is related to the binding energy of NH$_4^+ \cdot$ A. Further, $\Delta V_{50}$ is converted to the kinetic energy of NH$_4^+ \cdot$ A in the center of mass (i.e., KE$_{cm,50}$) using a parameterization of mass-dependent ion-mobility (Zaytsev et al. (2019) and details in the Supplement S3). In this way, TE$_B$ is related to a measurable parameter KE$_{cm,50}$. The mathematical relationship between TE$_B$ and KE$_{cm,50}$, TE$_B$ = f(KE$_{cm,50}$), is the final component to constrain the sensitivity.

We utilize the extensive calibration of 60 compounds from diverse chemical classes to derive the relationship between TE$_B$ and KE$_{cm,50}$. By rearranging Eqns. 13 and 14, and representing $\int_0^t [NH_4^+ \cdot H_2O]\, dt$ as a constant C, TE$_B$ can be expressed as Eqn. 15, where S$_{corr}$ represents the sensitivity corrected for f$_{NH_4^+ \cdot A}$, k, and TE$_\frac{m}{Q}$. Using Eqn. 15, the relationship between TE$_B$ and KE$_{cm,50}$ can be obtained through plotting S$_{corr}$ against KE$_{cm,50}$. As shown in Figure 5, S$_{corr}$ exhibits a positive dependence on KE$_{cm,50}$. The relationship between S$_{corr}$ and KE$_{cm,50}$ of the majority of compounds can be reasonably described using a

Hill Equation. Analytes with small KE$_{cm,50}$ (i.e., < 0.15 eV) have very low sensitivity, because of declustering of NH$_4^+ \cdot$ A in the electric fields. As KE$_{cm,50}$ increases, the sensitivity increases. This is because NH$_4^+ \cdot$ A with a stronger bond between A and NH$_4^+$ is more likely to survive the imparted energy from electric fields and hence more likely to be detected. When KE$_{cm,50}$ exceeds a threshold (i.e., 0.35 eV), NH$_4^+ \cdot$ A does not decluster in the electric field and it is detected with maximum S$_{corr}$. The maximum S$_{corr}$ is constrained using 2-hexanone here, but calibrations of analytes with KE$_{cm,50}$ larger than 0.35 eV are

warranted to constrain the maximum S$_{corr}$. Such analytes tend to be large oxygenated organic compounds with low volatility, making their calibrations challenging.

A similar relationship between S$_{corr}$ and KE$_{cm,50}$ has been reported in Zaytsev et al. (2019), which used a NH$_4^+$ – PTR3 and explored the relationship between the sensitivity and KE$_{cm,50}$ for 16 compounds, 9 of which are ketones. Unlike this study, Zaytsev et al. (2019) did not normalize the sensitivity to the ion-molecule collision rate constant k. This is reasonable as the

ion-molecule reaction time in NH$_4^+$ – PTR is ∼3 ms, about 15 times longer than that in our instrument. The long reaction time results in an equilibrium between cluster formation and fragmentation in the IMR for many analytes. In this thermodynamic regime, the product ion formation is proportional to the equilibrium constant of Eqn. 6 (Iyer et al., 2016; Robinson et al., 2022), so that normalizing the sensitivity in NH$_4^+$ – PTR3 by the equilibrium constant may improve its relationship with KE$_{cm,50}$.

In this study, the relationship between S$_{corr}$ and KE$_{cm,50}$ is largely defined by monofunctional organic compounds, but we

anticipate this relationship applies to organic compounds containing at least one functional group that binds strongly with NH$_4^+$, such as C=O, -OH, and nitrile. For example, five multifunctional compounds studied here (i.e., ethylene glycol, 1,3-propanediol, hydroxyacetone, 2,3-butanedione, and 2,3-pentanedione) are well described by the fitted Hill equation. Because





the fitted Hill equation does not apply to monocarboxylic acids, for reasons discussed later, the applicability of the relationship to multifunctional organic acids is uncertain and it warrants future investigation. Moreover, we compare several structural

isomers with monofunctional group, including acetone vs propanal, MACR vs MVK, C5-C9 mono-aldehyde vs mono-ketones. Despite the difference in $S_{corr}$ between isomers, their $S_{corr}$ and $KE_{cm,50}$ follow the same relationship.

The relationship between $S_{corr}$ and $KE_{cm,50}$ depicted in Figure 5 provides an effective approach to estimate the sensitivity of the $NH_4^+$ CIMS towards a suite of oxygenated organic compounds. The $KE_{cm,50}$ can be calculated from the voltage scan tests. $TE_{\frac{m}{Q}}$ can be experimentally quantified following the procedure in Section 2.2. $f_{NH_4^+ \cdot A}$ is unknown, but it is close to 1 for

multifunctional organic compounds, as discussed in Section 3.4. The k is also unknown, but it can be either calculated (Su, 1994) or reasonably estimated based on the molecular mass, elemental composition, and functional group (Sekimoto et al., 2017). k is generally on the order of $10^{-9}$ $cm^3$ $molecule^{-1}$ $s^{-1}$ (Figure S6). Finally, based on above-mentioned four parameters, the sensitivity can be estimated.

The observed relationship between $S_{corr}$ and $KE_{cm,50}$ in Figure 5 has limitations. First, it is only applicable to analytes of

which the ligand-switching reaction with $NH_4^+ \cdot H_2O$ is exothermic. This arises from approximating the ion-molecule reaction rate constant (k) in $S_{corr}$ using the collisional limiting rate constant. This approximation is not valid for endothermic reactions, which occur at a slower rate. This likely explains why several compounds, including monocarboxylic acids, some monoterpenes, reduced aromatics, isoprene, and 2-methylfuran, are outliers in Figure 5. For example, the $NH_4^+$ affinity of acetic acid is estimated to be lower than $H_2O$ (Section S5). Two monoterpenes, limonene and α-pinene, do not follow the fitted line, but the

behaviors of monoterpenes are more complicated. The calculated $NH_4^+$ affinities of β-pinene and camphene are smaller than that of $H_2O$ (Table S2), causing their ligand-switching reactions to be endothermic, but they fall on the fitted Hill equation. In contrast, limonene has larger $NH_4^+$ affinity than $H_2O$, but it is lower than the fitted line. The reason for such different behavior is unknown, but might be related to their structural difference. For example, β-pinene and camphene have an external C=C bond connected to the six-member ring, but α-pinene and limonene do not. Another limitation is that $KE_{cm,50}$, which is calculated

from $\Delta V_{50}$ based on voltage scan, may not be a proper proxy of $NH_4^+$ affinity for some analytes. For example, α-pinene has similar $NH_4^+$ affinity as β-pinene and camphene (Canaval et al., 2019), but the voltage scan test shows that α-pinene has a larger $KE_{cm,50}$ than the other two (Figure 5). Another exception is that isoprene and 2-methylfuran are expected to have small $NH_4^+$ affinity, considering their low sensitivities, but their $KE_{cm,50}$ is the highest among all analytes studied here. Similar "false positive" behavior (i.e., large $KE_{cm,50}$ or binding energy, but low sensitivity) is also observed in the $I^-$ CIMS (Iyer et al., 2016).

We suspect the voltage scanning affects not only the collisional energy of the $NH_4^+ \cdot A$, but also the ion-molecule chemistry or ion transmission via some unknown mechanisms. In the voltage scan, the FIMR front voltage is increased simultaneously with FIMR back voltage to keep the upstream voltage gradient constant. It is generally assumed that the absolute voltages do not affect the ion-molecule chemistry and transmission, as long as the voltage gradient is constant, but this assumption may not be valid. For example, in the voltage scan, we observe that the signal of reagent ion becomes noisy when the FIMR front voltage

(450 V) is close to the ion source voltage (440 V), suggesting that the FIMR front voltage affects the ion transmission from the ion source into the FIMR.





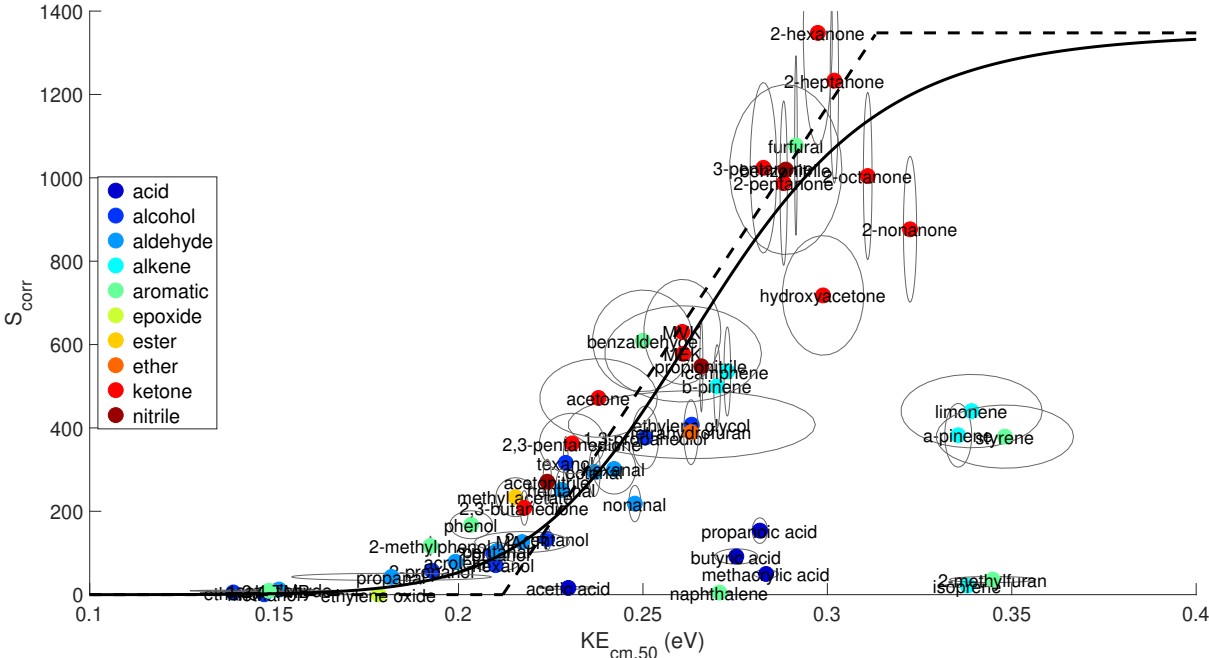

**Figure 5.** Relationship between $S_{corr}$ and $KE_{cm,50}$. $S_{corr}$ represents the sensitivity (cps ppbv$^{-1}$) corrected for the fraction of parent ion in all product ions ($f_{NH_4^+ \cdot A}$), m/Q-dependent transmission efficiency ($TE_{m/Q}$), and the ion-molecule reaction rate constant (k, $10^{-9}$ cm$^3$ molecule$^{-1}$), as defined in Eqn. 15. The solid line represents a fitting of analytes using a Hill Equation. $S_{corr} = 1350/(1+(0.267/KE_{cm})^{11})$. The dashed line represents a linear fitting for analytes with $KE_{cm}$ between 0.2 and 0.3 eV in a similar fashion done in Zaytsev et al. (2019). Organic acids, naphthalene, isoprene, 2-methyl furan, limonene, α-pinene, and styrene are excluded from both fittings. The ellipses represent the uncertainty range.

### 3.6 Comparison of Sensitivities between Instruments

In this section, we compare the sensitivities of our NH$_4^+$ CIMS (denoted as NOAA NH$_4^+$ CIMS) to two other NH$_4^+$ CIMS and a PTR using H$_3$O$^+$ chemistry. The other two NH$_4^+$ CIMS include a PTR3 instrument with a different IMR design from our Vocus
(Zaytsev et al., 2019) (denoted as PTR3 NH$_4^+$ CIMS) and a Vocus instrument with the same IMR design as ours but operated under different conditions (Khare et al., 2022) (denoted as Khare NH$_4^+$ CIMS). The PTR instrument is from our lab (denoted NOAA H$_3$O$^+$ CIMS), which uses the same FIMR as our NH$_4^+$ CIMS and was calibrated along with our NH$_4^+$ CIMS using the same calibration methods. The sensitivities of PTR3 NH$_4^+$ CIMS and Khare NH$_4^+$ CIMS are obtained from the corresponding references.

Figure 6 shows the sensitivity ratio of a selected instrument (S$_i$) to the NOAA NH$_4^+$ CIMS (S$_{NH_4^+ CIMS}$) for a number of analytes grouped by their chemical class. Khare NH$_4^+$ CIMS used the same ion source and IMR as NOAA NH$_4^+$ CIMS, but the sensitivities are generally lower than NOAA NH$_4^+$ CIMS by a factor of 5. In particular, the ethylene glycol sensitivity is lower by a factor of 100. The lower sensitivity in Khare et al. (2022) is likely because they used a higher NH$_3$/H$_2$O ratio than





this study. Khare et al. (2022) used 1 sccm vapor from a 1% ammonium hydroxide solution, while this study used 1 sccm
vapor from a 0.5% solution. As discussed in Section 3.2, larger $NH_3/H_2O$ ratio leads to a larger fraction of $NH_4^+ \cdot NH_3$ in the
total reagent ions and hence reduced sensitivity for most analytes (Figure 3). The sensitivities in Khare et al. (2022) can be
reproduced in NOAA $NH_4^+$ CIMS by using a larger $NH_3$ flow rate. The comparison between NOAA $NH_4^+$ CIMS and Khare
$NH_4^+$ CIMS further emphasizes the importance of FIMR conditions on the instrument performance.

The sensitivity ratio of NOAA $H_3O^+$ CIMS to NOAA $NH_4^+$ CIMS does spans a wide range from 1 to $10^4$. In general, the
sensitivity ratio anti-correlates with the sensitivity of NOAA $NH_4^+$ CIMS within each chemical class. This trend is the most
evident for aromatics. For example, for reduced aromatics, of which the sensitivities are smaller than 2 cps ppbv$^{-1}$ in the
NOAA $NH_4^+$ CIMS, their sensitivities are $10^3$ higher in the NOAA $H_3O^+$ CIMS. However, for oxygenated aromatics, such as
benzaldehyde and furfural, of which the sensitivities are on the order of $10^3$ cps ppbv$^{-1}$ in NOAA $NH_4^+$ CIMS, two instruments
have similar sensitivities. Therefore, $H_3O^+$ chemistry is more suitable to quantify reduced VOCs and small oxygenated VOCs
(e.g., acetic acid, methanol, acetaldehyde) than $NH_4^+ \cdot H_2O$ chemistry. $NH_4^+ \cdot H_2O$ chemistry is better for quantifying larger
oxygenated VOCs, because it causes less fragmentation than the $H_3O^+$ chemistry, which simplifies the interpretation of the
mass spectra.

Using the same $NH_4^+ \cdot H_2O$ chemistry, the PTR3 sensitivities are overall 20 times higher than those of NOAA $NH_4^+$ CIMS.
This difference is mainly due to different designs of the IMR and ion source. The PTR3 utilized a tripole electrode as IMR
(Breitenlechner et al., 2017). This design enables the IMR to be operated at 60 mbar and 3 ms reaction time (Zaytsev et al.,
2019), which are much higher than 3 mbar and 0.2 ms in NOAA $NH_4^+$ CIMS, and leads to enhanced sensitivities. The NOAA
$NH_4^+$ CIMS utilizes a low pressure discharge ion source, which generates more ions than the corona discharge ion source in the
PTR3. This compensates the effects of the lower IMR pressure and short reaction reaction on sensitivity to some extent. The
combined influences of ion source, IMR pressure, and reaction time result in the difference in sensitivities between NOAA $NH_4^+$
CIMS and PTR3 $NH_4^+$ CIMS. Despite of lower sensitivities, one advantage of the NOAA $NH_4^+$ CIMS is that its sensitivities
have much smaller dependence on the sample relative humidity that the PTR3 $NH_4^+$ CIMS does (Zaytsev et al., 2019).

## 4    Field Deployment

The $NH_4^+$ CIMS was deployed during the RECAP campaign in Pasadena, California in August-September, 2021. Measurements
presented in this section were made from August 10th to 19th when the instrument continuously sampled gas phase.

### 4.1    Measurement Capability

Figure S7 uses a mass defect plot to illustrate the measurement capability of $NH_4^+$ CIMS. In the RECAP campaign, a total
of 288 ions have signals above the detection limit. Half of the ions have the formula $C_xH_yN_1O_z$ (reagent ion included in the
formula). These ions mostly represent the non-nitrogen-containing oxygenated organics cluster with $NH_4^+$ or $NH_4^+ \cdot H_2O$. 70
ions have the formula $C_xH_yN_2O_z$, which likely represent nitrogen-containing compounds. This assignment is supported by the


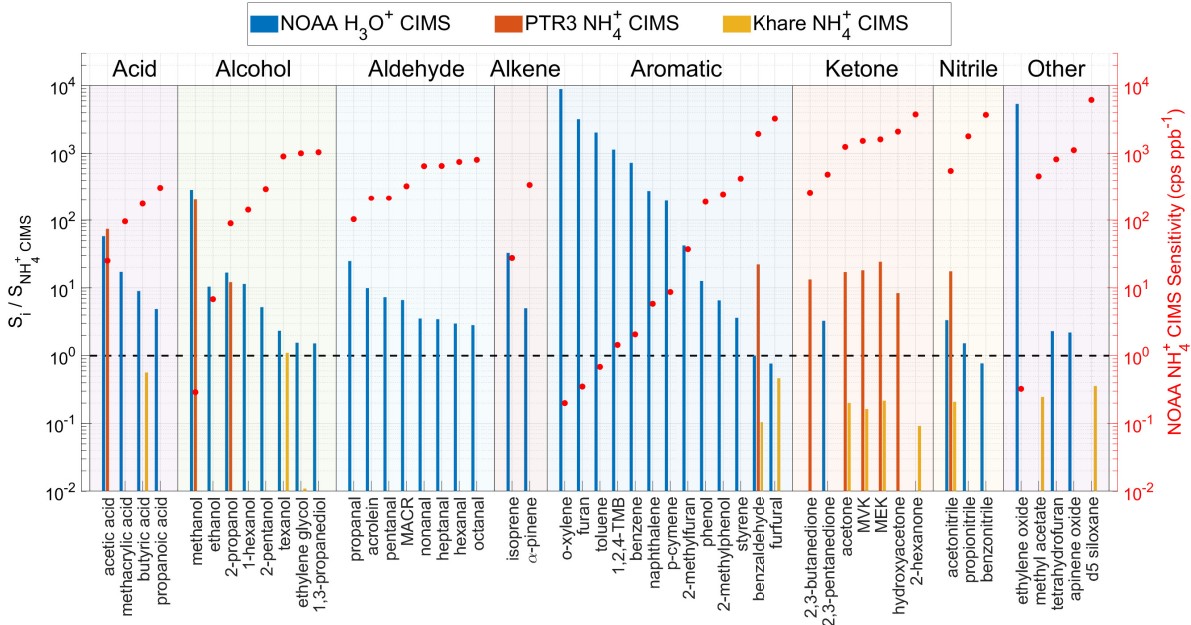

**Figure 6.** The sensitivity ratio of a selected instrument ($S_i$) to the NOAA $NH_4^+$ CIMS ($S_{NH_4^+ \, CIMS}$). The selected instrument i includes PTR3 $NH_4^+$ CIMS, Khare $NH_4^+$ CIMS, and NOAA $H_3O^+$ CIMS. The analytes are grouped by their chemical class. Within each chemical class, the analytes are sorted by their sensitivity.

analysis of product distribution (Section 3.4), which shows the product ion contains at most one nitrogen from the reagent ion. 40 out of 288 ions have the formula $C_xH_yO_z$, which likely represent analytes clustering with $H^+ \cdot (H_2O)_n$ (n=0,1,2).

### 4.2  Instrument Intercomparison

The co-located instruments in the RECAP campaign enable the evaluation of the field performance of $NH_4^+$ CIMS. In this section, we compare the measurements of several important atmospheric species from different chemical classes by 4 mass

spectrometers, $NH_4^+$ CIMS, $H_3O^+$ CIMS (Coggon et al., In prep.), $I^-$ CIMS (Robinson et al., 2022), and GC-MS (Gilman et al., 2015). For compounds that are commercially available, we calibrate the instrumental sensitivity and compare the mixing ratio. For multifunctional oxygenated organics for which no calibration standards exist, such as hydroxy nitrates, raw signals are compared. If multiple isomers exist for a parent ion and if these isomers are quantified by GC-MS, we apply the GC-MS resolved isomer ratio and the sensitivities of individual isomers to convert the raw cps of the parent ion to the summed mixing

ratio of all isomers for $NH_4^+$ CIMS (Supplement S7).

To account for instrument variability, the ion signals are typically normalized to the changing reagent ion signals. However, previous studies using Vocus in $H_3O^+$ and $NH_4^+ \cdot H_2O$ chemistry did not normalize the signals to reagent ions (Krechmer et al., 2018; Khare et al., 2022), because the BSQ serves as a high-pass band filter and substantially reduces the signal intensity




of reagent ions. In this study, we find that without normalization, the comparisons between $NH_4^+$ CIMS and GC-MS exhibit
significant difference between day and night (Figure S8a and b), which is consistent with the diurnal trend of reagent ion
$NH_4^+ \cdot H_2O$ (Figure S8c). Normalization to the reagent ion signal largely eliminates this difference. In light of this observation,
we normalize the ion signals to that of $NH_4^+ \cdot H_2O$ and then apply the normalized sensitivity to convert the signal (ncps) to
mixing ratio (ppbv).

### 4.2.1 Reduced VOCs

We compare the measurements of isoprene and monoterpenes between $NH_4^+$ CIMS, $H_3O^+$ CIMS, and GC-MS (Figure 7a
and b). $NH_4^+$ CIMS has a relatively low sensitivity towards isoprene (i.e., 28 cps ppbv$^{-1}$), but the high mass resolution of the
instrument enables a clear separation of isoprene (detected as $NH_4^+ \cdot C_5H_8$) from other isobars. Overall, isoprene measured by
$NH_4^+$ CIMS is ~20% higher than $H_3O^+$ CIMS and GC-MS (Figure S9). At night, both $NH_4^+$ CIMS and $H_3O^+$ CIMS observe
significantly higher isoprene concentration than GC-MS does (Figure 7a). This is likely because isoprene measured by $NH_4^+$
CIMS and $H_3O^+$ CIMS has interference from fragments of other species. Coggon et al. (In prep.) found that several aldehydes,
including octanal and nonanal, fragment in the $H_3O^+$ CIMS and produces $H^+ \cdot C_5H_8$. Correcting such interference results in
lower isoprene concentration measured by the $H_3O^+$ CIMS, particularly at night, and better agreement between $H_3O^+$ CIMS
and GC-MS (Figure 7a). Similarly, pentanal in the $NH_4^+$ CIMS produces $NH_4^+ \cdot C_5H_8$, which is the parent ion of isoprene.
Because the isoprene sensitivity in $NH_4^+$ CIMS is so low, the production of $NH_4^+ \cdot C_5H_8$ from an analyte with high sensitivity
would lead to large interference in isoprene concentration. Thus, $NH_4^+$ CIMS is not recommended for quantifying isoprene.

For monoterpenes, GC-MS shows that α-pinene and β-pinene are the dominant monoterpene isomers at the sampling site.
The ratio of α-pinene and β-pinene measured by GC-MS is used to convert $NH_4^+ \cdot C_{10}H_{16}$ signal measured by $NH_4^+$ CIMS to the
mixing ratio of total monoterpenes (Supplement S7). Three instruments show a large difference in measuring monoterpenes
(Figure 7b). The correlation between $H_3O^+$ CIMS and $NH_4^+$ CIMS is strong, but $H_3O^+$ CIMS observes three times more
monoterpenes than $NH_4^+$ CIMS (Figure S9b). $NH_4^+$ CIMS and GC-MS agree well at night, but $NH_4^+$ CIMS detects more
monoterpenes in the afternoon than GC-MS does (Figure 7b). The monoterpenes concentrations measured by both the $NH_4^+$
CIMS and the $H_3O^+$ CIMS are very spiky in the afternoon and the afternoon peak in the diurnal trend coincides with that of
isoprene (Figure S11). Both observations suggest that the monoterpenes are primary emissions from a local source, which is
likely the trees a few meters away from the sampling site. The absent of an afternoon peak of monoterpenes in GC-MS may be
because there are monoterpene isomers, other than α-pinene and β-pinene, which are not reported by the GC-MS. We do not
find evidence of fragmentation interference in monoterpenes in both $NH_4^+$ CIMS and the $H_3O^+$ CIMS.

### 4.2.2 Carbonyls

Figure 7c shows the time series of acetone measured by $NH_4^+$ CIMS, $H_3O^+$ CIMS, and GC-MS. In $NH_4^+$ CIMS, we attribute
the $NH_4^+ \cdot C_3H_6O$ solely to acetone and ignore the contribution from its structural isomer propanal, because GC-MS shows
propanal concentration is much lower than acetone and because the $NH_4^+$ CIMS sensitivity towards acetone is 10 times larger
than propanal (1247 vs 103 cps ppbv$^{-1}$). Acetone concentrations measured by the three instruments agree within 30%, which





is within the combined calibration uncertainties. Similar to acetone, the MEK measurement agrees between $NH_4^+$ CIMS and GC-MS within 20% and an $r^2$ of 0.91 (Figure 7d).

For MACR+MVK, three instruments agree well in the day, but $NH_4^+$ CIMS and $H_3O^+$ CIMS observe higher concentration
of MACR+MVK at night than GC-MS does (Figure 7e). We suspect the nighttime signal measured by $NH_4^+$ CIMS and $H_3O^+$ CIMS is due to 2-butenal and 3-butenal from cooking emissions.

### 4.2.3  Hydroxy nitrates

As shown in Figure S7, $NH_4^+$ CIMS detects a number of organic nitrates. Due to a lack of calibration standards, the sensitivities of organic nitrates in the $NH_4^+$ CIMS have not been quantified. Here, we explore the measurement capability of the $NH_4^+$ CIMS
by comparing the measurements of three organic nitrates between the $NH_4^+$ CIMS and the $I^-$ CIMS, $C_4H_7NO_5$, $C_5H_9NO_4$, and $C_{10}H_{17}NO_4$, all of which are detected as adduct ions in both instruments.

$C_4H_7NO_5$ matches the formula of hydroxynitrates produced from the oxidation of MACR+MVK. The $NH_4^+$ CIMS and the $I^-$ CIMS show a remarkably strong correlation with an $r^2$ value of 0.97 (Figure 8a and S10a). $C_4H_7NO_5$ corresponds to at least two structural isomers, one from the oxidation of MACR and the other from MVK. They have three functional groups (-OH,
-C=O, and $-ONO_2$). The strong correlation between two instruments could be due to a dominance of a single isomer or similar sensitivity toward both isomers in each instrument.

For $C_5H_9NO_4$, which has been attributed to isoprene-derived hydroxy nitrates in the literature (Lee et al., 2016; Xiong et al., 2015; Nguyen et al., 2015), the correlation $r^2$ between two measurements is only 0.54 (Figure 8b and S10b). The $C_5H_9NO_4$ measured by $I^-$ CIMS is close to zero at night, consistent with the isoprene-derived hydroxy nitrates previously measured
by the $CF_3O^-$ CIMS at the same site in 2017 (Vasquez et al., 2020). In contrast, the $C_5H_9NO_4$ measured by $NH_4^+$ CIMS is persistently high throughout a day (Figure 8b). We hypothesize that the $C_5H_9NO_4$ signal measured by the $NH_4^+$ has a large contribution from nitrooxy ketones, which are produced from the oxidation of pentenes by nitrate radical (Figure S12). Based on the laboratory characterization, $NH_4^+ \cdot H_2O$ is more sensitive to ketones than alcohols (Table 1 and Figure 5). Thus, it is possible that nitrooxy ketones from pentene oxidation have a much higher sensitivity than isoprene hydroxy nitrates in the
$NH_4^+$ CIMS. This leads to that the observed $C_5H_9NO_4$ signal in the $NH_4^+$ CIMS largely arises from nitrooxy ketones, even though their concentrations are smaller than isoprene hydroxy nitrates. In contrast, the $I^-$ CIMS is likely more sensitive to hydroxy nitrates than nitrooxy ketones (Lee et al., 2014). Further, the nighttime signal of $C_5H_9NO_4$ measured by the $NH_4^+$ CIMS is consistent with the observation that pentenes peak at night (Figure S13).

Lastly, we compare the measurements of $C_{10}H_{17}NO_4$, which represent the monoterpenes-derived hydroxy nitrates. Given
that there are at least four structural isomers of $C_{10}H_{17}NO_4$ (Xu et al., 2019), the agreement between two instruments is reasonable (Figure 8c), with $r^2$ equal to 0.75 (Figure S10c).



## 5 Conclusions

In this study, we describe the development and deployment of a CIMS using $NH_4^+ \cdot H_2O$ as reagent ion. $NH_4^+ \cdot H_2O$ is a highly versatile reagent ion for measurements of a wide range of oxygenated organic compounds. The instrument sensitivities and

product distributions are strongly dependent on the instrument conditions, including FIMR reduced electric field, temperature, pressure, the $H_2O$ mixing ratio, and the ratio of $NH_3$ to $H_2O$. These conditions should be carefully selected to ensure $NH_4^+ \cdot H_2O$ as the predominant reagent ion and to optimize sensitivities. For example, a comparison between this study and another study using the same instrument but under different FIMR conditions shows that the instrument sensitivity can differ by a factor of 5. Besides the desired reagent ion $NH_4^+ \cdot H_2O$, several other reagent ions exist in the FIMR even at the optimal condition, which

complicates the ion-molecule chemistry and the product distribution. The cluster ion $NH_4^+ \cdot A$ is the predominant product ion for acids, ketones, nitriles, and multifunctional oxygenated compounds. More diverse products, including protonated ion $H^+ \cdot A$ and fragmentation ions, are observed for small alcohols, biogenic VOCs, and reduced aromatics.

For monofunctional analytes, the $NH_4^+ \cdot H_2O$ chemistry exhibits high sensitivity (i.e., > 1000 cps ppbv$^{-1}$) towards ketones, moderate sensitivity (i.e., between 100 and 1000 cps ppbv$^{-1}$) towards aldehdyes, alcohols, organic acids, and monoterpenes,

low sensitivity (i.e., between 10 and 100 cps ppbv$^{-1}$) towards isoprene and C1 and C2 organics, and negligible sensitivity (i.e., < 10 cps ppbv$^{-1}$) towards reduced aromatics. The sensitivity of the $NH_4^+$ CIMS towards organic nitrates and highly oxygenated compounds requires further investigation. Overall, the $NH_4^+$ CIMS is complementary to existing chemical ionization schemes. Comparing to two commonly used reagent ions $H_3O^+$ and $I^-$, $NH_4^+ \cdot H_2O$ is more suitable to quantify moderately oxygenated compounds with one or two functional groups (i.e., C=O, -OH, and nitrile). These types of compounds have relatively low

sensitivity in $I^-$ CIMS (Lee et al., 2014). $H_3O^+$ and $NH_4^+ \cdot H_2O$ show similar sensitivity towards the moderately oxygenated compounds, and one advantage of $NH_4^+ \cdot H_2O$ chemistry is that it causes less fragmentation than $H_3O^+$ chemistry, which simplifies the interpretation of the mass spectra. Moreover, we reveal a strong relationship between instrumental sensitivity and the binding energy of the analyte-$NH_4^+$ cluster, which can be estimated using voltage scanning tests. This offers the possibility to constrain the sensitivity of analytes for which no calibration standards exist.

The field performance of the $NH_4^+$ CIMS is evaluated based on comparisons with three co-located mass spectrometers in the RECAP campaign during a 10-day period. $NH_4^+$ CIMS and GC-MS show reasonable agreement in measuring carbonyls (i.e., acetone, MEK, MACR+MVK), but not in isoprene and monoterpenes. Isoprene measured by the $NH_4^+$ CIMS has fragmentation interference. The difference in monoterpene measurements is possible because some monoterpene isomers are not reported by the GC-MS. A number of nitrogen-containing species are detected by the $NH_4^+$ CIMS and three representative ones

are compared to $I^-$ CIMS. Strong correlations are observed for $C_4H_7NO_5$ (likely oxidation products of MACR and MVK) and $C_{10}H_{17}NO_4$ (likely oxidation products of monoterpenes), but not for $C_5H_9NO_4$ (including isoprene hydroxy nitrates and nitrooxy ketones from pentene oxidation). The difference in $C_5H_9NO_4$ measurements is likely because $NH_4^+$ CIMS and $I^-$ CIMS have vastly different sensitivities toward different structural isomers. Such comparisons illustrate the unique measurement capability of the $NH_4^+$ CIMS, which is complementary to existing chemical ionization schemes.





**Figure 7.** The time series and diurnal trend of selected species measured by NH$_4^+$ CIMS, H$_3$O$^+$ CIMS, and GC-MS. (a) Isoprene; (b) Monterpenes; (c) Acetone; (d) Methyl Ethyl Ketone (MEK); (e) Methacrolein (MACR) + Methyl Vinyl Ketone (MVK). In panel (a), H$_3$O$^+$ CIMS corr represents the isoprene measurement by the H$_3$O$^+$ CIMS after correcting the interference from octanal and nonanal (Supplement S7). In panel (d), MEK measured by the NOAA H$_3$O$^+$ CIMS is not included because its peak fitting (C$_4$H$_9$O$^+$) is degraded by the nearby large signal of H$_9$O$_4^+$.

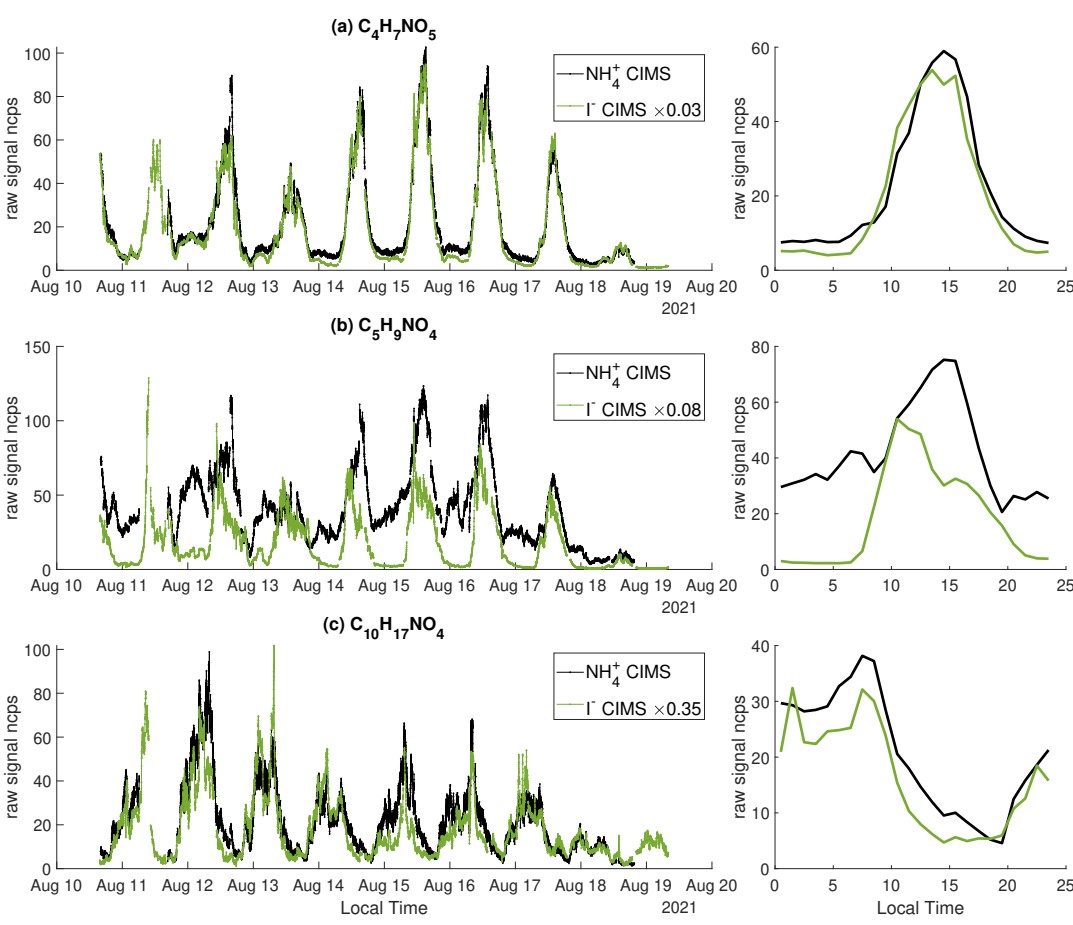

**Figure 8.** The time series and diurnal trend of three nitrogen-containing species measured by $NH_4^+$ CIMS and $I^-$ CIMS. (a) $C_4H_7NO_5$; (b) $C_5H_9NO_4$; (c) $C_{10}H_{17}NO_4$. Because of a lack of calibration standards, the raw signals (ncps) are shown here.





*Data availability.* Data from the RECAP campaign are available to the general public at
https://csl.noaa.gov/groups/csl7/measurements/2021sunvex/GroundLA/DataDownload/

*Author contributions.* LX and CW designed the research, LX operated the $NH_4^+$ CIMS, MMC and CES operated the $H_3O^+$ CIMS, JBG and AL operated the GC-MS, MAR, JAN, and PRV operated the $I^-$ CIMS, MB, GAN, MMC, CES, and SSB provided critical support and comments in $NH_4^+$ CIMS operation, and KHM performed theoretical calculations. All authors commented on the manuscript.

*Competing interests.* The authors declare that they have no competing interests.

*Acknowledgements.* We thank Paul Wennberg and John Crounse for their support in RECAP campaign. We thank Kristian H. Møller and Henrik G. Kjaergaard for calculation of dipole moments and polarizabilities. This work was supported by the NOAA Cooperative Agreement with CIRES, NA17OAR4320101. The NOAA Chemical Sciences Laboratory acknowledges support for this work from the California Air Resources Board under agreement number 20RD002.



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
