# Peer review of "Chemical Ionization Mass Spectrometry Utilizing Ammonium Ions (NH4+ CIMS) for Measurements of Organic Compounds in the Atmosphere"

_Atmospheric Measurement Techniques, 2022_

## Author Comment (AC1)

We thank the reviewers for their helpful comments. We reply to the reviewers' comments point by point as indicated below. The reviewers' comments are in italics and changes made to the manuscript are in quotation marks.

**Reviewer #1**

*The manuscript presents a detailed overview of the operation and sensitivity of NH4+ chemical ionization applied in Vocus-type mass spectrometer that uses an improved ion-molecule reactor FIMR. The authors discuss operational constraints, sensitivity calibration as a function of cluster stability (inferred from dissociation voltage inside the instrument) and present detailed comparison of selected species with other CIMS types in the field. The paper is thorough in presenting limitations and advantages of this chemical ionization technique and therefore it will be very valuable for the community. Some comments and suggestions are outlined below.*

*General comments*

*While I appreciate how detailed the paper is, I found it quite long as it includes a lot of information: investigations into different instrumental parameters, sensitivity comparison to other instruments as well as CIMS intercomparison in the field. I found few repetitive parts of text that could be shortened to my mind, both between Methods and Results (section 4), and in Supplement. I suggest authors go through the text to try to make paragraphs shorter and focus clearer. I also found myself jumping between formulas in different parts of main text and Supplementary to get the full picture.*

Authors Reply: We thank the reviewer for the positive comment. As the manuscript focuses on instrument development and characterization, we strive to provide sufficient details for others to evaluate and replicate our study. We agree with the reviewer that the broad aspects and detailed discussions prolong the manuscript. To increase the readability, we have made the following changes. All the line numbers below refer to the original manuscript.

(1) Consolidate all experimental methods into section 2.2 and then simplify this section by moving less important details into the Supplement. The changes include moving the procedure to quantify product distribution from section 3.4 (Lines 281-286) to section 2.2; consolidating the description of voltage scan tests by moving some discussions from section 3.5 (Lines 342 - 348) to section 2.2;

moving the description about standard gas cylinders and liquid calibration unit from section 2.2 to the Supplement.

(2) Remove the repetitive definitions of sensitivity unit. (Lines 47, 320, 321)

(3) Remove the repetitive description of voltage scan tests. (Lines 342 - 348)

(4) Remove the repetitive description of PTR modification (Lines 131 - 132)

(5) Remove the repetitive definition of FIMR conditions (Line 203).

*Specific comments*

*1. Line 36: Please include Lindinger et al. (1998) and Blake et al. (2006) as references to early NH4+ ionization works.*

Authors Reply: These two references have been added.

*2. Lines 46 and 49: Maybe for comparison, report sensitivities in same unit as in your calibrations, cps/ppb?*

Authors Reply: The sensitivity unit has been consistently expressed as cps ppbv$^{-1}$ throughout the revised manuscript.

*3. Equation 10: would be nice to report the value of µ0 used.*

Authors Reply: The values of $\mu_0$ have been added to Table S1 in the revised manuscript.

*4. Section 3.3. in relation to section 3.2: It would be useful to see the modelled reagent ion distribution for the conditions that were selected for sensitivity characterization? T = 314K, P = 3mbar was used in Section 3.3., was it the same for determining sensitivities in Table 1? It could be mentioned in Methods.*

Authors Reply: The sensitivities in Table 1 are determined under FIMR drift voltage 55 Td, FIMR pressure 3 mbar, FIMR temperature 314 K, 1 sccm from 0.5% ammonium hydroxide aqueous solution, and 20 sccm water vapor, as stated in lines 269 – 271 in the original manuscript.

The modeled reagent ion distribution under above conditions are shown in Figure R1.

[Figure]

Figure R1. Modeled distribution of reagent ions using the experimental conditions.

*5. Line 253-254: Authors say that different compounds have maximum sensitivity at different ratios of reagent ions because they have different reactivities "towards NH4+.H2O and other reagent ions". What does this mean? Isn't it so that sensitivity is calculated based only on A.NH4+ and the sensitivity is a function of the cluster stability?*

Authors Reply: Yes, the sensitivity is calculated based only on A•NH$_4^+$ and the sensitivity depends on the cluster stability. However, A•NH$_4^+$ can be produced by reactions of analyte A with different reagent ions (i.e., NH$_4^+$•H$_2$O and NH$_4^+$•NH$_3$). Analytes have different reactivities towards NH$_4^+$•H$_2$O and NH$_4^+$•NH$_3$ and such difference varies among analytes. One analyte may only react with NH$_4^+$•H$_2$O, but not NH$_4^+$•NH$_3$, while another analyte may have similar reactivity towards NH$_4^+$•H$_2$O and NH$_4^+$•NH$_3$. Therefore, the maximum sensitivity of different analytes occurs at different reagent ion distribution (i.e., NH$_4^+$•H$_2$O/H$_3$O$^+$•H$_2$O ratio and NH$_4^+$•H$_2$O/NH$_4^+$•NH$_3$ ratio).

*6. Figure 3: Line colors are so that it is hard to see. Could the compounds be grouped somehow like in Fig. 5, if it is feasible and improves the visibility of lines. How is optimal range determined? By eye, even smaller ratios would be okay.*

Authors Reply: The rationale behind the current layout in Figure 3 and Figure S2 is to compare the behavior of all analytes studied here in an integrated fashion. We agree with the reviewer that the sheer number of tracers worsens the plot readability. In the revised manuscript, we break the

figure into 6 panels by grouping compounds based on their chemical functional classes. The revised figures are shown below.

[Figure]

The optimal range is determined in a somewhat arbitrary way in the original manuscript. To be more precise, the optimal range should be determined based on analytes of interest. Thus, we have removed the "recommended range" from the figure and rephrase the sentence as follows:

"In summary, the instrument sensitivities should be calibrated as a function of $NH_4^+ \cdot H_2O/H_3O^+ \cdot H_2O$ ratio. Then, the optimal $NH_4^+ \cdot H_2O/H_3O^+ \cdot H_2O$ range can be selected based on the analytes of interest."

*7. lines 217-218: Authors say that E/N affects 3 things: reagent ion distribution, focusing effect and declustering. I think in FIMR E/N affects declustering and therefore this regulated reagent ion distribution, so one leads from another. The authors mention just this in the beginning of this paragraph.*

Authors Reply: We agree with the reviewer that reagent ion distribution and the extent of declustering are coupled. We have rephrased the sentence as follows:

"Overall, the observed dependence of sensitivities on E/N is a superposition of at least three effects, focusing effects and the extent of declustering of both reagent ions and product ions."

*8. Lines 343-346: This already is described in section 2.1. It could be more concise here. And also same text is again in Supplement, section 3.*

Authors Reply: We have revised the manuscript accordingly to make the discussions more concise.

*9. Figure 5: Could authors specify somewhere why KEcm,50 is different than in Zaytsev et al. 2019? I assume it is the geometry of the ion-molecule reactor?*

Authors Reply: The difference in $KE_{cm,50}$ between this study and Zaytsev et al. is caused by the assumed distance between FIMR back and skimmer (i.e., the parameter d in Eqn. S6). This distance is assumed to be 3 mm in this study and it is not specified in Zaytsev et al.

*10. Line 425: Could authors clarify what "anticorrelation" mean here? Orange data points and right axis on Figure 6 are absolute CIMS sensitivity and left axis (blue bars) is ratio, so if NOAA H3O+ would have constant sensitivity, I think we would observe similar trend in the ratio as now.*

Authors Reply: The "anti-correlation" means that the sensitivity ratio of $H_3O^+$ CIMS to $NH_4^+$ CIMS ($S_{H3O+}/S_{NH4+}$) decreases as the absolute sensitivity of $NH_4^+$ CIMS increases. Take the aromatics as an example. The $S_{H3O+}/S_{NH4+}$ is $10^4$ for o-xylene, which has very low sensitivity in

the $NH_4^+$ CIMS (i.e., < 1 cps/ppb). In contrast, the $S_{H3O+}/S_{NH4+}$ decreases to 1 for benzaldehyde, which has high sensitivity in the $NH_4^+$ CIMS (i.e., > 1000 cps/ppb). Here, we emphasize the change in the absolute sensitivity of the $NH_4^+$ CIMS, not the $H_3O^+$ CIMS. The $NH_4^+$ CIMS sensitivity spans a much wider range than the $H_3O^+$ CIMS.

*11. Lines 430-432: A statement that is not necessarily a conclusion from the data presented in this paper. I would suggest to include a reference or remove. Also lines 543-544.*

Authors Reply: We believe the statement that $NH_4^+ \cdot H_2O$ chemistry is better for quantifying larger oxygenated VOCs than $H_3O^+$ chemistry is valid, because $NH_4^+ \cdot H_2O$ causes less fragmentation than the $H_3O^+$ chemistry. Several references discussing the fragmentation of oxygenated organics in the PTR have been added in the revised manuscript (Pagonis et al., 2019; Sekimoto et al., 2017; Yuan et al., 2017).

*12. Lines 440-441: Similar as above. Where is it shown in the paper how sensitivity in the presented instrument changes (or doesn't) with RH? Maybe in Methods, line 104, it is mentioned. Would it be possible to add a figure to SI?*

Authors Reply: A figure showing the minimal dependence of sensitivity on the sample RH is shown below (Figure R2) and has been added to the revised manuscript. A similar observation was shown in a recent study by Khare et al. (2022).

[Figure]

Figure R2. Dependence of instrument sensitivities of representative species on sample RH.

*13. Line 455: Which H3O+ CIMS is this? NOAA?*

Authors Reply: It is the NOAA $H_3O^+$ CIMS. This has been specified in the revised manuscript.

*14. Line 523: It seems to me that in Figure S13, pentenes peak in early morning.*

Authors Reply: We agree with the reviewer that pentenes peak in early morning as shown in Figure S13. The main point is not when the concentrations of pentenes peak, but their concentrations are higher at night than in the day. It is also important to note that the formation of nitrooxy ketones depends on not only pentenes concentration, but also the concentration of nitrate radicals. To be more precise, we have rephrased the sentence to the following in the revised manuscript.

"Further, the nighttime signal of $C_5H_9NO_4$ measured by the $NH_4^+$ CIMS is consistent with the observation that the pentenes are higher at night than in the day."

*15. Line 547: "strong relationship"- maybe worth mentioning exceptions here (like monoterpenes and other).*

Authors Reply: In light of reviewer's suggestion, we have added the following sentence in the revised manuscript.

"Caution is required when applying this method, because the observed relationship is only applicable to analytes of which the ligand-switching reaction with $NH_4^+ \cdot H_2O$ is exothermic and because the measured $KE_{cm,50}$ may not be a proper proxy of $NH_4^+$ affinity for some analytes. The combination of experimental constraints and theoretical calculation of analyte thermodynamic properties could potentially provide more accurate estimate of analyte sensitivities."

*16. Equation S6: Are "E" in Eqn. S6 and "E" Eqn. 10 same variables?*

Authors Reply: The parameter E in both equations represents the electric field strength, but it refers to different regions. The E in Eqn. S6 refers to the voltage gradient between FIMR back and skimmer. The E in Eqn. 10 refers to that between FIMR front and back. To clarify, we have added subscript to E in Eqn. S6 (i.e., $E_{FIMR\ back\ -\ skimmer}$).

*17. Figure S7: If possible, it would be very useful if authors included a table with detected ion compositions in this figure.*

Authors Reply: This is a great suggestion and we will upload the Tofwerk peak list as a supplement file.

*Technical corrections*

*1. Line 66: Change "Tofwer" to "Tofwerk".*

Authors Reply: Done.

*2. Line 146: Remove word "does" after H2O.*

Authors Reply: Done.

*3. Line 164: Authors use I+ in here probably with "I" standing for "ion", so H3O+ and NH4+.*
*It could be mentioned in the text, just like A is NH3 and H2O.*

Authors Reply: "I" has been defined as ion in the line above this sentence.

*4. Figure 1: Somewhat difficult to separate solid and dashed lines in the legend.*

Authors Reply: We have modified the figure legend and the update figure is shown below.

[Figure]

*5. Line 323: Here, f(NH4+.H2O) is defined, but it is already used in previous section.*

Authors Reply: We prefer to reminding the reader of the $f_{NH4+\cdot H2O}$ definition here, because the previous definition and the last appearance of $f_{NH4+\cdot H2O}$ is too far away.

*6. Line 333: Change "k have less uncertain" to "k is less uncertain".*

Authors Reply: Done.

*7. Line 476: Change "produces" to "produce".*

Authors Reply: Done.

*8. Line 489: Change "absent" to "absence".*

Authors Reply: Done.

*9. Line 506: Suggestion to change "as adduct ions" to "as adducts with reagent ions".*

Authors Reply: Done.

*10. Line 516: Change "a day" to "the day".*

Authors Reply: Done.

*11. Supplementary line 58: Change "dipoment" to "dipole moment".*

Authors Reply: Done.

*This paper describes the optimization and calibration of a commercially available Chemical Ionization Time of Flight Mass Spectrometer for the measurement of a suite of organic compounds in the atmosphere. Field data from the instrument and comparisons to other instrumentation are also presented. The instrument utilizes NH4+ reagent ions and a relatively new (less than 5 year old design) Focusing Ion Molecule Reactor (FIMR). The authors present the advantages and limitations of using NH4+ chemistry. The paper is very detailed (more on this later) providing more than enough information for other groups to reproduce the results presented here. It is well suited for publication in AMT and will be of great value to the community. That said, I do have a few comments and suggestions below.*

*General Comments:*

*As mentioned above the paper is very detailed. While this is a good thing, there are times this borders on overwhelming and does make the paper quite long. At times I wondered if could be split in two, with the field measurements and instrument intercomparisons being their own paper. The sheer number of species quantified (or estimated using voltage scanning) by the instrument also makes some of the figures difficult to digest. Figure 3 is an example of this. While I understand what the authors are trying to show, the sheer number of traces makes it almost impossible pick out individual species. Anyone suffering color blindness would have no clue what they are looking at. At times, I also found it difficult follow through the jumping around of formulas in the modelling of the reagent ion distribution. Again, I appreciate the level of detail the authors have provided the readers of this paper, I simply feel it should be tightened up in places and presented in a more digestible way.*

Authors Reply: We thank the reviewer for the positive comment. As the manuscript focuses on instrument development and characterization, we strive to provide sufficient details for others to evaluate and replicate our study. We agree with the reviewer that the broad aspects and detailed discussions prolong the manuscript. To increase the readability, we have removed some repetitive definitions, consolidated all experimental methods into a single section, and made the discussions more concise. Please see detailed changes above in our response to reviewer#1's general comment. To increase figure readability, Figure 3 and Figure S2 in the original manuscript have been broken down into 6 panels by grouping compounds based on their chemical functional classes. We prefer

to keep the instrument characterization, intercomparison, and field deployment in a single manuscript in order to provide a coherent, complete, and comprehensive story.

*Specific Comments*

*1. Title: I assume the authors meant "A Chemical Ionization Mass Spectrometer Utilizing....." or "Chemical Ionization Mass Spectrometry Utilizing....."*

Authors Reply: We thank the reviewer for the suggestion. The manuscript title has been changed to "Chemical Ionization Mass Spectrometry Utilizing Ammonium Ions ($NH_4^+$ CIMS) for Measurements of Organic Compounds in the Atmosphere"

*2. P3 L66: Tofwerk Vocus*

Authors Reply: Done.

*3. P11 Figure 2: A plot of the model results showing the final optimal conditions would be nice to see here somewhere.*

Authors Reply: Quantitatively modeling the dependence of instrument sensitivities on the FIMR conditions is desirable but challenging. It is because the model omits the focusing effects and does not accurately represent the energetic collisions caused by accelerating voltages. Moreover, the $NH_4^+$ affinity of analytes and other thermodynamics of reactions between analytes and reagent ions are limited and uncertain in the literature.

*4. P13 Figure 3: As mentioned above this is a very busy plot. I'm wondering if there's a way the species could be grouped or if the authors could split the plot into panels so that if would be easier to see what is going on. Right now it kind of looks like a mess.*

Authors Reply: The figure has been modified accordingly. The revised figure is shown above in response to comment#6 of reviewer #1.

*5. P14 L281: We have performed……..and measured……*

Authors Reply: Done.

*6. P15 L308: oxygenated aromatics have a…….*

Authors Reply: Done.

*7. P16 L333: …..k have less uncertainty than TE.*

Authors Reply: We have modified it to "k is less uncertain than TE".

*8. P16 L333: You really should not start a sentence with a lower case variable. Perhaps "The value of k......"*

Authors Reply: Done.

*9. P17 L343: This has been mentioned elsewhere and is an opportunity to tighten up the manuscript.*

Authors Reply: We have revised the manuscript accordingly.

*10. P21 Figure 5: I'm finding it difficult to wrap my head around the sensitivity comparisons between the instruments without taking into account the change in reagent ion signal (sensitivities are cps/ppb not ncps/ppbv). Fluctuations in reagent ions will surely occur. The authors even mention the example of day vs night. I realize this is difficult in the VOCUS instrument since proper counting of the reagent ions puts significant wear on the MCP detectors, but I'm wondering if there is some other way to account for this.*

Authors Reply: We believe this comment refers to Figure 6, instead of Figure 5. The sensitivities (unit cps/ppb) are obtained from laboratory calibrations where the fluctuations in reagent ions are minimal. As the reviewer pointed out, the proper counting of the reagent ions in the VOCUS is challenging and we do not have a proper way to account for the variation in reagent ion signal in field measurements. We are working to correct for this by constantly introducing a deuterated analyte into the system to track fluctuations in the instrument, though this was not performed during this study.

*11. P22 L476: should be "produce"*

Authors Reply: Done.

*12. P22 L489: Absence?*

Authors Reply: Done.

*13. P23 L516: ....throughout the day.*

Authors Reply: Done.

*14. P3 L58: "Further, the dipole moment....."*

Authors Reply: Done.

*15. P4 Figure S2: Same comments as Figure 3.*

Authors Reply: The same change as Figure 3 has been applied to Figure S2. The revised figures are shown below.

[Figure]

***Other Changes***

1. After the submission of the manuscript, there are some minor changes to the GC-MS data. The updated data are now included in the revised manuscript. The changes in GC-MS data lead to better agreement between the $NH_4^+$ CIMS, $H_3O^+$ CIMS, and GC-MS. For example, the acetone concentrations measured by the three instruments now agree within 15%, which was 30% in the original manuscript.

Reference

Khare, P., Krechmer, J. E., Machesky, J. E., Hass-Mitchell, T., Cao, C., Wang, J., Majluf, F., Lopez-Hilfiker, F., Malek, S., Wang, W., Seltzer, K., Pye, H. O. T., Commane, R., McDonald, B. C., Toledo-Crow, R., Mak, J. E., and Gentner, D. R.: Ammonium-adduct chemical ionization to investigate anthropogenic oxygenated gas-phase organic compounds in urban air, Atmos. Chem. Phys. Discuss., 2022, 1-39, 10.5194/acp-2022-421, 2022.

Pagonis, D., Sekimoto, K., and de Gouw, J.: A Library of Proton-Transfer Reactions of H3O+ Ions Used for Trace Gas Detection, J Am Soc Mass Spectr, 30, 1330-1335, 10.1007/s13361-019-02209-3, 2019.

Sekimoto, K., Li, S.-M., Yuan, B., Koss, A., Coggon, M., Warneke, C., and de Gouw, J.: Calculation of the sensitivity of proton-transfer-reaction mass spectrometry (PTR-MS) for organic trace gases using molecular properties, Int J Mass Spectrom, 421, 71-94, https://doi.org/10.1016/j.ijms.2017.04.006, 2017.

Yuan, B., Koss, A. R., Warneke, C., Coggon, M., Sekimoto, K., and de Gouw, J. A.: Proton-Transfer-Reaction Mass Spectrometry: Applications in Atmospheric Sciences, Chem Rev, 117, 13187-13229, 10.1021/acs.chemrev.7b00325, 2017.